# Imaging analysis of six human histone H1 variants reveals universal enrichment of H1.2, H1.3, and H1.5 at the nuclear periphery and nucleolar H1X presence

**Monica Salinas-Pena, Elena Rebollo, Albert Jordan***

Molecular Biology Institute of Barcelona (IBMB-CSIC), Barcelona, Spain

**Abstract** Histone H1 participates in chromatin condensation and regulates nuclear processes. Human somatic cells may contain up to seven histone H1 variants, although their functional heterogeneity is not fully understood. Here, we have profiled the differential nuclear distribution of the somatic H1 repertoire in human cells through imaging techniques including super-resolution microscopy. H1 variants exhibit characteristic distribution patterns in both interphase and mitosis. H1.2, H1.3, and H1.5 are universally enriched at the nuclear periphery in all cell lines analyzed and co-localize with compacted DNA. H1.0 shows a less pronounced peripheral localization, with apparent variability among different cell lines. On the other hand, H1.4 and H1X are distributed throughout the nucleus, being H1X universally enriched in high-GC regions and abundant in the nucleoli. Interestingly, H1.4 and H1.0 show a more peripheral distribution in cell lines lacking H1.3 and H1.5. The differential distribution patterns of H1 suggest specific functionalities in organizing lamina-associated domains or nucleolar activity, which is further supported by a distinct response of H1X or phosphorylated H1.4 to the inhibition of ribosomal DNA transcription. Moreover, H1 variants depletion affects chromatin structure in a variant-specific manner. Concretely, H1.2 knock-down, either alone or combined, triggers a global chromatin decompaction. Overall, imaging has allowed us to distinguish H1 variants distribution beyond the segregation in two groups denoted by previous ChIP-Seq determinations. Our results support H1 variants heterogeneity and suggest that variant-specific functionality can be shared between different cell types.

***For correspondence:**
ajvbmc@ibmb.csic.es

**Competing interest:** The authors declare that no competing interests exist.

## eLife assessment

This manuscript is an **important** advance in the study of Histone H1s, finding distinct distributions of various H1 variants in the genome. The controls presented by the authors provide **convincing** evidence to demonstrate a heterogenous distribution of H1 which might reflect functional regulation of chromatin accessibility by linker histones. This work will be of interest to the genome organization field, and could additionally provide a framework for understanding H1 mis-regulation observed in cancer cells.

## Introduction

The eukaryotic genome is organized in a functional and spatially segregated manner within the interphase nucleus. Nuclear architecture plays a crucial role in gene regulation and defining cellular identity. Early studies led to the classical view of a bipartite chromatin composition in which euchromatin is defined as active chromatin located at the nuclear center while heterochromatin corresponds to the more compact chromatin fraction, generally positioned at the nuclear periphery and surrounding the

nucleoli (*Cremer and Cremer, 2001*; *Jost et al., 2012*; *Solovei et al., 2016*). While heterochromatin correlates with late-replicating, low-GC chromatin, euchromatin is characterized by an early replication timing and high-GC content. This differential epigenetic landscape of the interphase nucleus can be also recapitulated by Giemsa bands (G-bands) (*Serna-Pujol et al., 2021*), which arise from the characteristic banding of metaphase chromosomes.

Chromatin organization involves several hierarchical levels, including chromosome territories (*Cremer and Cremer, 2001*; *Girelli et al., 2020*), A (active) and B (inactive) compartments at the megabase level (*Lieberman-Aiden et al., 2009*), topological associated domains (TADs) (*Dixon et al., 2012*; *Nora et al., 2012*; *Sexton et al., 2012*), and chromatin loops. Moreover, chromatin segregation is facilitated by chromatin tethering to scaffolding structures. This anchorage originates nuclear environments referred as lamina-associated domains (LADs) or nucleolus-associated domains (NADs), which constitute heterochromatic regions anchored to the nuclear lamina and the nucleolus, respectively.

Human cells have 1000–1500 LADs that cover more than one-third of the genome (*Guelen et al., 2008*). LADs represent a well-known repressive environment, characterized by low gene density, low gene expression, and a great overlap with B compartment. LADs are enriched in repressive histone modifications, including H3K9me2, which is considered a conserved chromatin mark of LADs (*Poleshko et al., 2019*). While H3K9me2 plays a part in anchoring chromatin to the nuclear lamina, current data indicate that it is probably not a sufficient signal in mammals, as other anchoring mechanisms may exist (*Harr et al., 2015*; *Kind et al., 2013*).

The nucleolus is a membraneless structure where ribosome biogenesis and regulation occurs. Nucleoli also act as central chromatin organizers. Genomic regions positioned close to nucleolus are referred as NADs, which were firstly genome-wide identified in human using a biochemical purification of nucleoli (*Dillinger et al., 2017*; *Németh et al., 2010*). NADs consist of mainly heterochromatic regions and an important overlap with LADs was found. Accordingly, some LADs have been found to stochastically reshuffled after mitosis and associate with nucleoli (*Kind et al., 2013*). These observations suggest that the lamina and nucleolus could act as interchangeable scaffolds for heterochromatin positioning. More recently, the inclusion of HiC-based approaches has provided more accurate genome-wide NADs maps (*Bersaglieri et al., 2022*; *Peng et al., 2023*).

Histone composition, including histone variants and their modifications, also plays a role in defining chromatin functionality (*Martire and Banaszynski, 2020*). In particular, linker histone H1 family is evolutionary diverse and human somatic cells may contain up to seven H1 variants (H1.1 to H1.5, H1.0, and H1X). Although H1 has classically been regarded as a general repressor, increasing evidence support H1 variants functional diversity in chromatin regulation (*Fyodorov et al., 2018*; *Millán-Ariño et al., 2016*). A compromised H1 content causes chromatin structural defects, evidenced both in human (*Serna-Pujol et al., 2022*) and mice models (*Geeven et al., 2015*; *Willcockson et al., 2021*; *Yusufova et al., 2021*). Although in these scenarios multiple H1 variants depletion leads to chromatin decompaction, the contribution of individual H1 variants in maintaining chromatin structure has not been studied.

A long-standing enigma concerning H1 is whether its variants have a uniform distribution in different cell types or, conversely, display cell-line-specific binding patterns. The presumption that H1 variants are specifically distributed among different cell lines comes from combining various pieces of evidence from different publications (*Cao et al., 2013*; *Izzo et al., 2013*; *Li et al., 2012*; *Millán-Ariño et al., 2014*; *Torres et al., 2016*). Nevertheless, no study has properly addressed the question up to date. Consequently, the absence of existing reports using standardized experimental and analytical workflows may lead to potential misinterpretation of the data. The only studies that performed a systematic analysis of different variants have been performed in a single-cell model and often used overexpression strategies. These include the analysis of H1.1–H1.5 using DamID in IMR-90 cells (*Izzo et al., 2013*), and a second report in which ChIP-Seq of endogenous H1.2 and H1X and the exogenous H1.0-HA and H1.4-HA was performed in T47D cells (*Millán-Ariño et al., 2014*). Recently, we performed the first genome-wide analysis of six endogenous H1 variants in human cells, bypassing the everlasting limitation of mapping exogenous proteins (*Serna-Pujol et al., 2022* and in preparation for H1.3 data). In T47D cells, H1 variants are distributed in two large groups depending on the local GC content. H1.2, H1.3, H1.5, and H1.0 are enriched at low-GC regions and B compartment while H1.4 and H1X are more abundant within high-GC regions and

A compartment. However, whether these distribution profiles are conserved among different cell types remains unknown.

H1 complement (i.e. H1 variants and its proportions present in a specific cell) is dynamic throughout differentiation and cancer and also varies between cell types. Whether these expression fluctuations translate into changes in distribution patterns has not been studied. The best-characterized example is H1.0, which accumulates during differentiation (*Terme et al., 2011*) and whose expression is associated to a less aggressive phenotype of tumoral cells (*Torres et al., 2016*). H1.0 has been described as a replacement histone, as it responds to a compromised H1 content. In breast cancer cells, combined depletion of H1.2 and H1.4 leads to H1.0 upregulation, although without significant redistribution alterations (*Izquierdo-Bouldstridge et al., 2017*; *Serna-Pujol et al., 2022*).

In this study, we provide novel insights into the differential nuclear distribution of somatic H1 variants in human cells, through an imaging approach. Super-resolution microscopy in T47D cells shows that H1.2, H1.3, H1.5 and, to a lesser extent, H1.0, are enriched at the nuclear periphery and coincide more with more compacted DNA, as supported by super-resolution microscopy. Contrarywise, H1X and H1.4 are distributed throughout the nucleus with a significant H1X enrichment in nucleoli. Differential distribution patterns suggest concrete implications in genome functionality and translate into variant-specific functional consequences upon H1 depletion. Specifically, single or combined H1.2 depletion triggers a general chromatin decompaction, which is not observed when depleting H1.4 or H1X. Furthermore, we conducted the first systematic comparison of six somatic H1 variants in several human cell lines, including ChIP-Seq profiling of H1X in different cell types, which has only been mapped in T47D breast cancer cells up to date. Interestingly, certain H1 variants display universal distribution patterns, despite variations in H1 complement across cell lines. H1.2, H1.3, and H1.5 are consistently enriched at the nuclear periphery while H1X is more abundant at high-GC regions and present at nucleoli in all cell lines evaluated. We identified a recurring concomitant absence of H1.3 and H1.5, and this specific H1 complement is associated with a more peripheral distribution of H1.0 and H1.4 proteins, suggesting potential compensatory mechanisms between variants. Altogether, our study represents a comprehensive attempt to systematically characterize the repertoire of somatic H1 variants, their differential distribution in the human genome and their functional diversity.

## Results

### Histone H1 variants are differentially enriched toward the periphery of the interphase nucleus and at nucleoli

We have recently reported that H1 variants are differentially distributed into two main groups within the genome of T47D cells: H1.0, H1.2, H1.3, and H1.5 are enriched in low-GC regions while H1.4 and H1X are more abundant at high-GC regions (*Serna-Pujol et al., 2022*) (hereafter referred as 'low-GC' or 'high-GC' H1 variants). However, how these variants are distributed along the nucleus, where chromatin is spatially arranged to regulate genome function, remains unknown. Immunofluorescence analysis demonstrated that different H1 variants exhibit unique nuclear patterns (*Figure 1A*). H1.2, H1.3, and H1.5 were observed to be enriched at the nuclear periphery, while H1.0 was distributed throughout the nucleus, with certain enrichment territories that tend to be peripheral. On the other hand, H1.4 and H1X were found to be homogeneously distributed throughout the nucleus, with the difference that H1X was particularly abundant in the nucleoli. H1.2, H1.3, H1.5, and H1.0 showed a coincident pattern with DNA staining one, suggesting an enrichment at more condensed-DNA nuclear areas, including but not limited to the nuclear periphery (*Figure 1A*, bottom panels). H1.4 profile only partially mimicked DNA pattern while H1X profile was opposite to that of DNA. Overall, these results are compatible with the classification of H1 variants into two differential groups, as previously suggested by ChIP-Seq analysis, as low-GC chromatin tends to be peripheral and coincides with late-replicating heterochromatin. Accordingly, co-immunostaining of H1 variants with HP1alpha denoted that low-GC H1s tend to better co-localize with this heterochromatin marker compared to high-GC H1 variants (*Figure 1—figure supplement 1C, D*).

To further examine H1 variants nuclear distribution, we performed an analysis of the H1 radial intensity distribution. Each nucleus was automatically divided into four sections of equal area (as exemplified in *Figure 1B*) and the percentage of H1 intensity present in each area was quantified (*Figure 1C*). H1.2, H1.3, and H1.5 showed a clear relationship with radiality, becoming increasingly

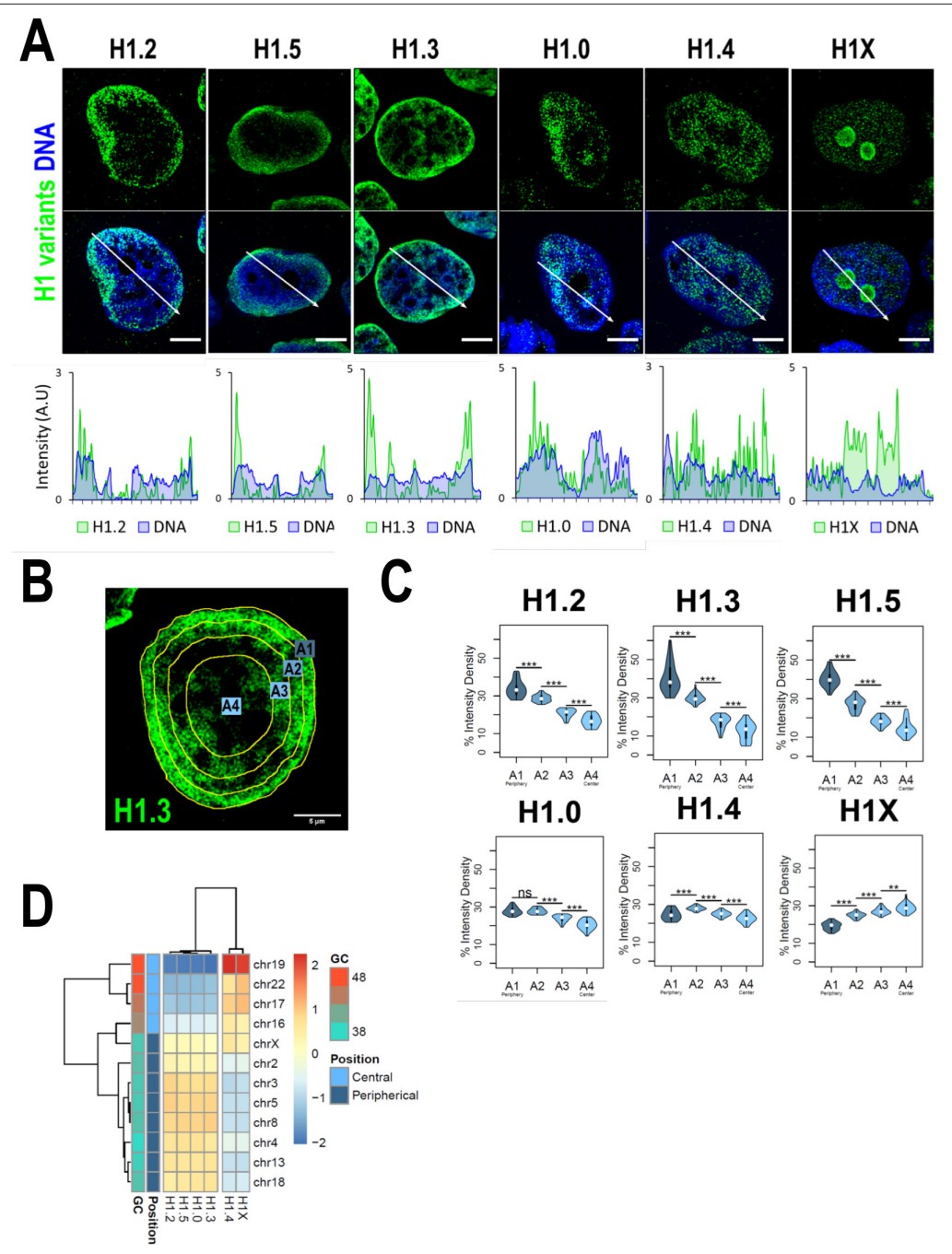

**Figure 1.** H1 variants are differentially enriched toward the nuclear periphery in T47D cells. (**A**) Confocal immunofluorescence of H1 variants (green) and DNA staining (blue). Bottom panels show the intensity profiles of H1 variants and DNA along the arrows depicted in the upper panel. Scale bar: 5 µm. (**B**) Example of one cell stained with H1.3 antibody in which four sections of an equivalent area and convergent to the nuclear center are shown. Sections are named A1–A4, from the more peripheral section to the more central one. H1 variants immunofluorescence intensity was measured in each area and expressed as percentage. (**C**) Quantifications of H1 variants using the segmentation illustrated in (**B**), where n = 30 cells/condition were quantified, and data were represented in violin plots. Statistical differences between A1–A2, A2–A3, and A3–A4 for H1.0 and H1.4 are supported by paired *t*-test. (***) p-value <0.001; (**) p-value <0.01; (ns/non-significant) p-value >0.05. (**D**) H1 variants Input-subtracted ChIP-Seq median abundance per chromosome. *Y*-axis annotation indicated median %GC content per chromosome and their nuclear positions according to *Boyle et al., 2001*; *Girelli et al., 2020*.

The online version of this article includes the following figure supplement(s) for figure 1:

*Figure 1 continued on next page*

*Figure 1 continued*

**Figure supplement 1.** Endogenous H1 variants immunofluorescence controls and co-localization with heterochromatin protein HP1.

abundant from the nuclear center to the periphery. Importantly, H1.0 was found to be most abundant at the two most peripheral percentiles. A different distribution was observed for H1.4, which was more equally distributed along A1–A4 sections, being more abundant at intermediate A2–A3 percentiles. H1X was gradually increasing toward the nucleus center. It is important to note that, in part, this gradual profile is due to the nucleolar H1X fraction, as nucleoli tend to be located at central nuclear positions and we are not excluding nucleoli from the analysis.

Differential H1 'radiality' is related to the spatial organization of chromatin in the nucleus of mammalian cells and the concept of chromosomes territories. Chromosomes are not randomly positioned in the nucleus; gene-poor chromosomes are located at peripheral positions while gene-rich chromosomes tend to occupy central regions (*Boyle et al., 2001*). We computed H1 variants ChIP-Seq abundance at chromosomes reported to occupy different radial territories (*Figure 1D*) and, supporting immunofluorescence quantification, we found that H1.2, H1.3, H1.5, and H1.0 were enriched at peripheral chromosomes over central ones. On the contrary, H1.4 and H1X were more abundant at chromosomes located in central positions.

## Super-resolution imaging shows that low-GC H1 variants co-localize with compacted DNA and do not overlap

We sought to extend the analysis to the super-resolution level. Super-resolution imaging techniques surpass the diffraction limit, enabling visualization of subcellular organization beyond conventional light microscopy resolution (≈250 nm). Specifically, we used super-resolution radial fluctuations (SRRF) technique (*Culley et al., 2018*; *Gustafsson et al., 2016*). Super-resolution imaging of H1 variants reinforced the different nuclear patterns already seen through confocal microscopy (*Figure 2A*). H1.2, H1.3, H1.5, and to a lesser extent H1.0 were specially detected at the nuclear periphery, but SRRF imaging emphasized their presence throughout the entire nucleus, highlighting that these variants are not limited to the nuclear periphery. For its part, H1.4 discrete signals were found throughout the whole nucleus, excluding nucleoli. Conversely, nucleolar H1X was detected but super-resolution accentuated the presence of the non-nucleolar H1X fraction, which was observed throughout the entire nucleus.

The percentage of co-localization of histone H1 variants and DNA signal detected by SRRF was calculated (*Figure 2B*). Of note, DNA super-resolution imaging distinguishes between areas of densely packed DNA and areas with little or no DNA signal, compared to the typically blurred DNA signal resolved by confocal resolution. Consequently, the DNA we detected through SRRF imaging represents chromatin in a more condensed or closed state, relative to surrounding regions. H1.2, H1.3, H1.5, and H1.0 showed a higher degree of co-localization with DNA compared to H1.4 and H1X, with H1X showing the least co-localization. In summary, our super-resolution co-localization studies of H1 variants with DNA reinforced the differential nuclear patterns obtained at a conventional resolution. Moreover, SRRF imaging highlights that, although showing preferential relative enrichment to concrete nuclear regions, such as nuclear periphery or nucleoli, H1 variants are not restricted to those compartments.

To further characterize H1 variants patterns, we next studied how H1.0 co-localizes with the rest of H1 variants in single nucleolus (*Figure 2C, D*, *Figure 2—figure supplement 1A–C*). Preferential co-localization of H1.0 with H1.2/H1.3/H1.5 over H1.4 and H1X was evident at confocal resolution (*Figure 2—figure supplement 1A–C*), suggesting that low-GC variants occupy similar nuclear territories and supporting ChIP-Seq profiling obtained when analyzing T47D cell population. However, at the super-resolution level all H1 variants co-localized poorly with H1.0, suggesting that they occupy different chromatin fibers (*Figure 2D*). This observation could be underlying the principles of nucleosome composition and 3D chromatin organization. Considering the dissimilarities observed between the standard and super-resolution approaches, results suggest that, in single cells, at least in more closed regions where H1.0/H1.2/H1.3/H1.5 are more abundant, H1 variants are not occupying random positions in nearby nucleosomes. If that were the case, co-localization between the different 'low-GC' variants (versus 'high-GC' ones) would not be lost when improving resolution. Therefore, a compatible

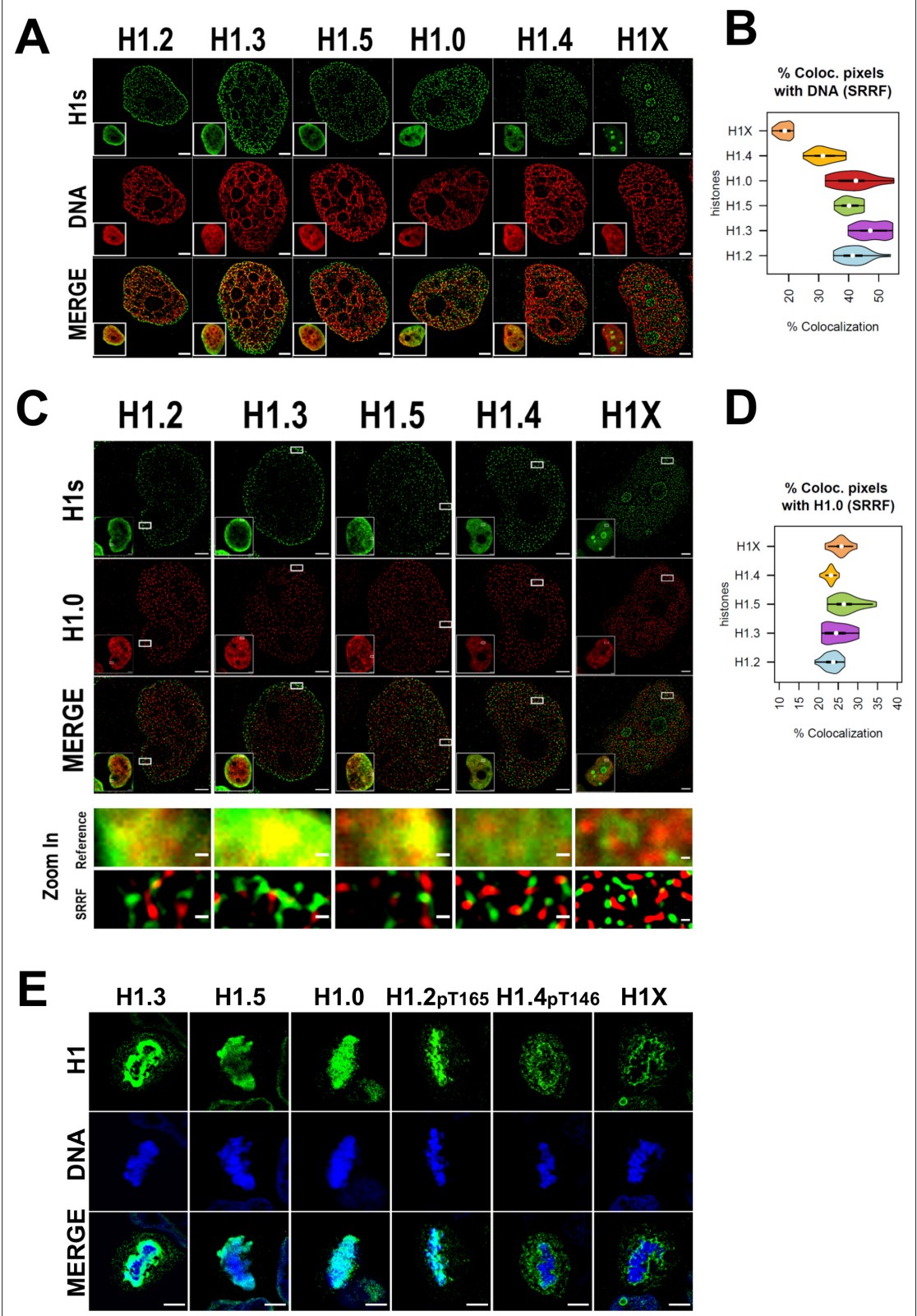

**Figure 2.** Super-resolution imaging shows that H1 variants occupy different regions in single cells. (**A**) Super-resolution radial fluctuation (SRRF) images of H1 variants (green) and DNA (red). Bottom-left: Insets show the Reference confocal image in each case. Scale bar: 2 μm. (**B**) Percentage of co-localized pixels between H1 variants and DNA by SRRF imaging. *n* = 20 cells/condition were quantified and values distribution were represented as violin plots. (**C**) SRRF images of H1 variants (green) and H1.0 (red). Bottom-left: Insets show the Reference confocal image in each case. In the bottom

*Figure 2 continued on next page*

*Figure 2 continued*

panel, the highlighted zoom-in insets at confocal (reference) or SRRF resolutions are shown. Scale bar: 2 µm, scale bar in zoom-in insets: 200 nm. (**D**) Percentage of co-localized pixels of H1 variants with H1.0 by SRRF imaging. *n* = 20 cells/condition were quantified and values distribution were represented as violin plots. (**E**) Immunofluorescence of H1 variants (green) and DNA (blue) during metaphase. As H1.2 and H1.4 signal was not detected during metaphase (see main text and *Figure 2—figure supplements 2 and 3*), antibodies recognizing specific phosphorylations of these variants were used. To see H1 variants profiles along mitosis progression, see *Figure 2—figure supplements 2 and 3*. Scale bar: 5 µm.

The online version of this article includes the following figure supplement(s) for figure 2:

**Figure supplement 1.** H1 variants co-localization at confocal resolution.

**Figure supplement 2.** H1 variants distribution patterns along mitosis phases.

**Figure supplement 3.** H1 variants phosphorylation during mitosis.

---

model with our results is that, in single cells, heterochromatic 3D nanodomains tend to be consistently marked by a certain H1 variant.

## Attachment of H1 to mitotic chromosomes differs between variants

As H1 variants showed different nuclear patterns in interphasic cells, we studied whether these differential patterns were also observed through mitosis (*Figure 2E*). Co-immunostaining of replication-independent H1.0 and H1X was performed and their distribution through consecutive phases of mitosis was monitored (*Figure 2—figure supplement 2A*). H1.0 and H1X exhibited completely different distribution patterns. While H1.0 was anchored to mitotic chromosomes, H1X was not recruited to mitotic chromosomes and it accumulated to the perichromosomal region.

H1 variants highly enriched at nuclear periphery during interphase were co-examined with Lamin A along mitotic progression (*Figure 2—figure supplement 2B–D*). Since nuclear lamina is disassembled during mitosis, we wondered whether the positional information of H1-marked chromatin was maintained through mitosis. H1.3 and H1.5 showed similar distribution profiles throughout mitosis (*Figure 2—figure supplement 2B, C*). Both H1 variants persisted at mitotic chromatin, being specially enriched at the periphery of condensed chromosomes. Importantly, H1.3 and H1.5 layers re-associate with the forming nuclear lamina before mitotic exit. Instead, H1.2 signal was dispersed after prophase (*Figure 2—figure supplement 2D*). H1.2 was re-detected at anaphase, when its peripheral enrichment was re-acquired, parallel to lamina re-assembly. These results suggest that radial position of H1.2-, H1.3-, and H1.5-marked chromatin is inherited through mitosis. These genomic regions are re-localized to the nuclear periphery following mitotic division and the nuclear lamina reassembles around H1.2-, H1.3-, or H1.5-associated chromatin.

The apparent absence of H1.2 (and H1.4, data not shown) at intermediate mitotic stages is striking. H1 proteins are highly phoshphorylated during mitosis, so we explored whether apparent mitotic absence was due to these H1 variants becoming highly post-translationally modified during mitosis, and the antibodies are not recognizing the H1-modified fraction. H1.2 phosphorylation of Threonine 165 (H1.2-pT165) and H1.4 phosphorylation of Threonine 146 (H1.4-pT146) were found to be highly increased in mitosis compared to interphase (*Figure 2—figure supplement 3A*). Thus, this confirms that H1.2 and H1.4 proteins are not absent at certain mitotic phases, but antibodies are unable to recognize their post-translationally modified state. Although these modifications are most prevalent in mitosis, they are also detected at interphase. Early-mitotic H1.2pT165 was found to be associated with condensed chromosomes, with maximum levels occurring at metaphase and drastically dropping down at later mitotic phases (*Figure 2—figure supplement 3B*). These temporal dynamics coincide with the previously discussed lack of H1.2 detection at metaphase and the re-appearance of H1.2 signal at anaphase/telophase (*Figure 2—figure supplement 2D*). Furthermore, H1.4-pT146 was also more enriched at early mitotic stages, but excluded from metaphasic chromosomes (*Figure 2—figure supplement 3C*). Instead, H1.4-pT146 was accumulated adjacent to chromosomes, in the perichromosomal layer, similar to what was observed for H1X (*Figure 2—figure supplement 2A*).

Overall, analysis of H1 variants during mitosis indicates that the two H1 groups defined by ChIP-Seq analysis present distinct localization patterns through mitosis. While 'low-GC' H1s (whether phosphorylated or not) are associated with mitotic chromosomes, 'high-GC' variants (phosphorylated or not) are excluded from mitotic chromosomes and accumulate to the perichromosomal region. In addition, H1.3 and H1.5 are enriched toward the peripheral chromosome regions, in comparison to H1.0 or

H1.2-pT165. In conclusion, imaging experiments support the differential distribution of H1 variants not only during interphase but also in mitotic cells.

## H1.2, H1.3, and H1.5 are enriched within LADs

Genome conformation is regulated by the tethering of chromatin to scaffold structures, such as the nuclear lamina or nucleolus. Several domains have been implicated in chromatin organization, such us LADs or NADs. Proper chromatin organization is crucial for genome functionality, so we further explored H1 variants differential distribution within these particular chromatin domains.

As H1.2, H1.3, and H1.5 were highly enriched at nuclear periphery (*Figures 1A and 2*) and re-associated to lamina before mitotic exit (*Figure 2—figure supplement 2B–D*), we aimed to explore their association with lamina through super-resolution microscopy (*Figure 3A*). Interestingly, in all three cases, the peripheral H1 enrichment seen by confocal microscopy was perfectly resolved by SRRF imaging as an H1 layer adjacent to Lamin A layer. We next performed H1 variants co-immunostaining with H3K9me2, an evolutionarily conserved specific mark of LADs (*Poleshko et al., 2019*). As expected, H3K9me2 was found enriched but not limited to nuclear periphery (*Figure 2—figure supplement 3A*). Indeed, H3K9me2 nuclear pattern resembled the distributions of H1.2/H1.3 and H1.5 and a high co-localization was observed at both confocal (*Figure 3B*, *Figure 3—figure supplement 1A*) and SRRF resolutions (*Figure 3C*).

As aforementioned, H1.0 was also found to be partially enriched at nuclear periphery (*Figure 1A, B*). Taking advantage of publicly available LADs coordinates, we computed H1 variants ChIP-Seq abundance in LADs (*Figure 3D*). 'Low-GC' H1 variants, including H1.0, were enriched at LADs in comparison to H1.4 and H1X. On the whole, we demonstrated that H1.2/H1.3/H1.5, and also H1.0, are constituents of lamina-associated chromatin, as supported by microscopic and ChIP-Seq experiments.

## H1X and phosphorylated H1.2 or H1.4 present differential nucleolar patterns

We previously identified H1X enriched at nucleolus, using both confocal and super-resolution microscopy (*Figures 1 and 2*). To begin with, nucleolar H1X enrichment was found using alternative permeabilization methods (data not shown) and both nucleolar and non-nucleolar signals were drastically reduced upon specific H1X depletion (*Figure 3—figure supplement 1B*), confirming that nucleolar H1X enrichment is not an artifact. Co-immunostaining of H1X and the nucleolar marker Nucleophosmin (NPM1) confirmed that H1X was located inside nucleoli, with a tendency to form a ring-like layer adjacent to NPM1 on the inner side of nucleoli (*Figure 3E, F*).

Nonetheless, H1X was not the only H1 variant present at nucleoli, as H1.2-pT165 and H1.4-pT146 were also present at nucleoli (*Figure 3F*, *Figure 2—figure supplement 3B, C*). While interphasic H1.2-pT165 was highly enriched within nucleoli, interphasic H1.4-pT146 was not restricted to nucleoli, as it was also detected within the rest of the nucleus, specially coinciding with DNA-free staining regions. H1.4-pT146 formed clusters of punctate nucleolar staining. These characteristic nucleolar dots seem to represent active ribosomal DNA (rDNA) transcription, resembling immunostaining patterns of active rDNA transcriptional machinery factors (i.e. UBF, RNApol I). A second interphasic H1.4-pT146 pattern was observed (*Figure 3—figure supplement 1C*), with no nucleolar enrichment but speckled enrichment territories along the nucleus that overlap with regions with less DNA staining, that is, less condensed chromatin. This speckled staining could represent the localization of transcriptionally active chromatin near RNA splicing factories, as has been proposed before for H1.4-pS187, which display an analogous interphasic staining (*Zheng et al., 2010*).

Next, we tested whether the nucleolar localization of H1 variants depends on nucleolar integrity, by treating cells with the rDNA transcription inhibitor Actinomycin D (ActD). ActD treatment triggers large-scale structural reorganization of the nucleoli, with the migration of some nucleolar markers to the nucleolar-remnant periphery, forming the so-called nucleolar caps while other nucleolar proteins are translocated to nucleoplasm (*Burger et al., 2010*). Each nucleolar cap represents UBF-loaded rDNA repeats from a single nucleolar organizer region (NOR). As expected, ActD treatment triggered a total translocation of NPM1 to nucleoplasm and alterations in DNA distribution were also evident (*Figure 3—figure supplement 1D*). H1.4-pT146 was redistributed to the nucleolar caps, as it would be expected from active rRNA transcription machinery components. Nucleolar enrichment of H1.2-pT165 was completely lost upon ActD treatment, and it was not detected at nucleolar caps. However,

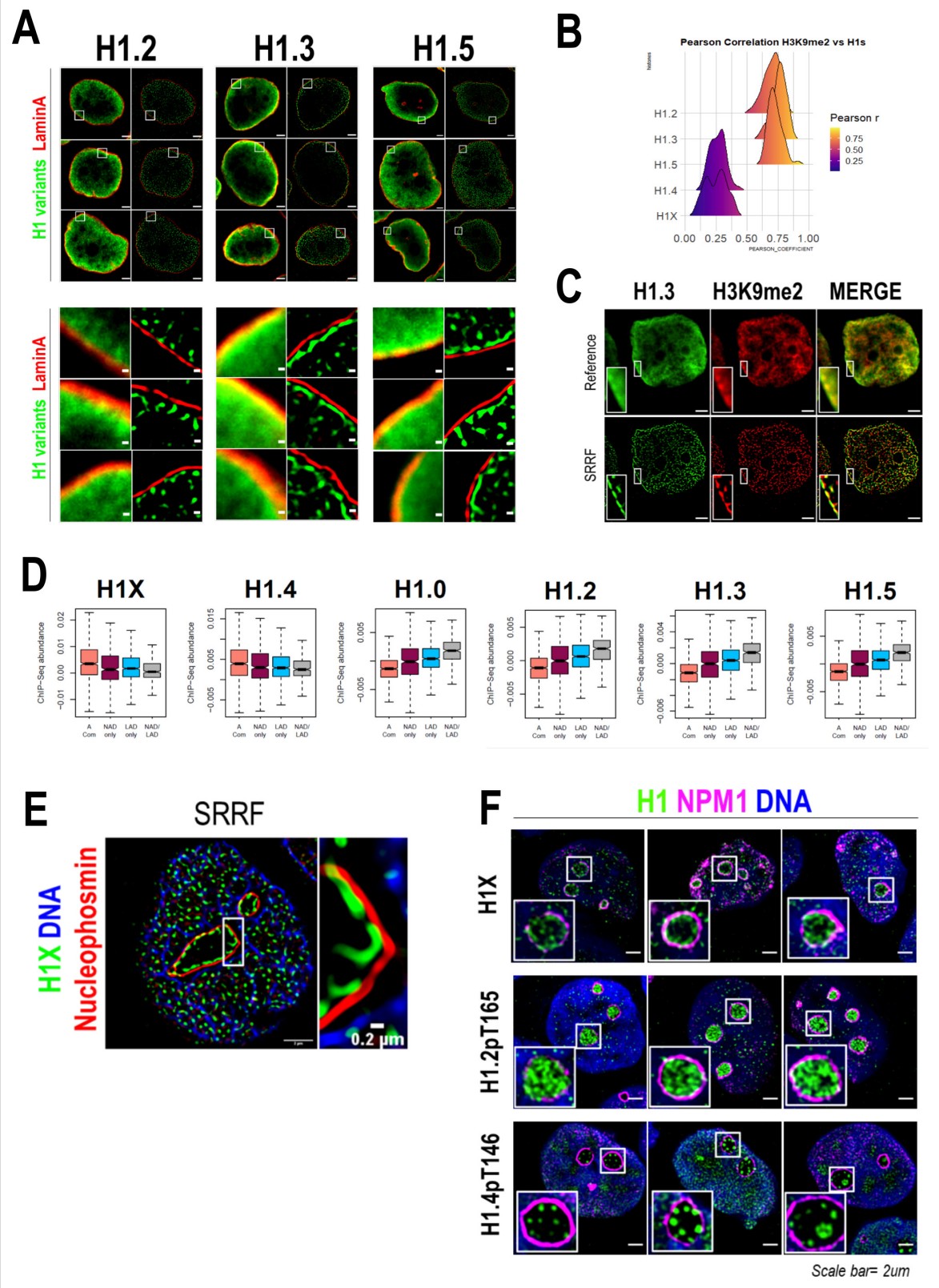

**Figure 3.** H1 variants presence within lamina-associated domains (LADs) and nucleoli. (**A**) Confocal (left) and super-resolution (right) images of a T47D cells stained for H1.2, H1.3, or H1.5 (in green) and Lamin A (in red) obtained using SRRF. Full nuclei (upper panel) and zoomed views of nuclear periphery (bottom panel) are shown. Scale bars: 2 μm (upper panel) and 200 nm (bottom panel). Three representative cells are shown for each H1 variant. (**B**) Pearson correlation coefficient (*r*) of H1 variants and H3K9me2 co-immunostaining signal. *r* values distribution in *n* = 50 cells/condition are shown.

*Figure 3 continued on next page*

*Figure 3 continued*

Representative immunofluorescence images of H1 variants and H3K9me2 are shown in *Figure 3—figure supplement 1A*. Scale bar: 5 μm. (**C**) H1.3 and H3K9me2 immunofluorescence at confocal (reference) and super-resolution (SRRF) level. A zoom-in inset of the peripheral layer formed by both H1.3 and H3K9me3 is shown. Scaler bar: 2 μm. (**D**) Boxplots show the Input-subtracted H1 variants ChIP-Seq abundance within regions exclusively defined as nucleolus-associated domains (NADs) (NAD only) or LADs (LAD only) and those genomic segments defined as both NADs and LADs (NAD/LAD). A compartment regions are included as a reference. NADs coordinates were extracted from *Peng et al., 2023*. (**E**) Representative SRRF image of H1X, NPM1, and DNA. Zoom-in highlights the H1X nucleolar layer. Scale bar: 2 μm. Scale bar in zoom-in: 0.2 μm. (**F**) Immunofluorescence of H1X, H1.2-pT165, or H1.4-pT146, Nucleophosmin (NPM1) and DNA. Insets show a zoom-in of a single nucleolus. Scale bar: 2 μm. Three representative cells are shown for each H1.

The online version of this article includes the following figure supplement(s) for figure 3:

**Figure supplement 1.** H1 variants within lamina-associated domains (LADs) and nucleoli under basal and upon ribosomal DNA (rDNA) transcription inhibition.

the H1X characteristic nucleolar ring was still found in the remnant nucleoli of a considerable fraction of cells. Thus, H1X and phosphorylated H1.2 or H1.4 exhibit characteristic nucleolar patterns, basally and upon ActD treatment, that could reflect different functional involvements in nucleolar dynamics.

We further explored H1 variants relationship with nucleolar organization by analyzing ChIP-Seq H1 variants abundance within NADs. We used a recently published NADs mapping performed in HeLa cells, which identified 264 NADs (*Peng et al., 2023*). As NADs and LADs show a substantial overlap, we analyzed separately those regions defined exclusively as NAD or LAD and those that overlap (NAD/LAD), similar to the analysis performed in *Bersaglieri et al., 2022*. 'Low-GC' H1 variants were enriched within both NADs and LADs, highlighting their presence within multiple repressive compartments in the nucleus. On the contrary, 'high-GC' H1 variants are depleted from both NADs and LADs repressive domains, compared to the A compartment (*Figure 3D*). Still, H1X and H1.4 are more

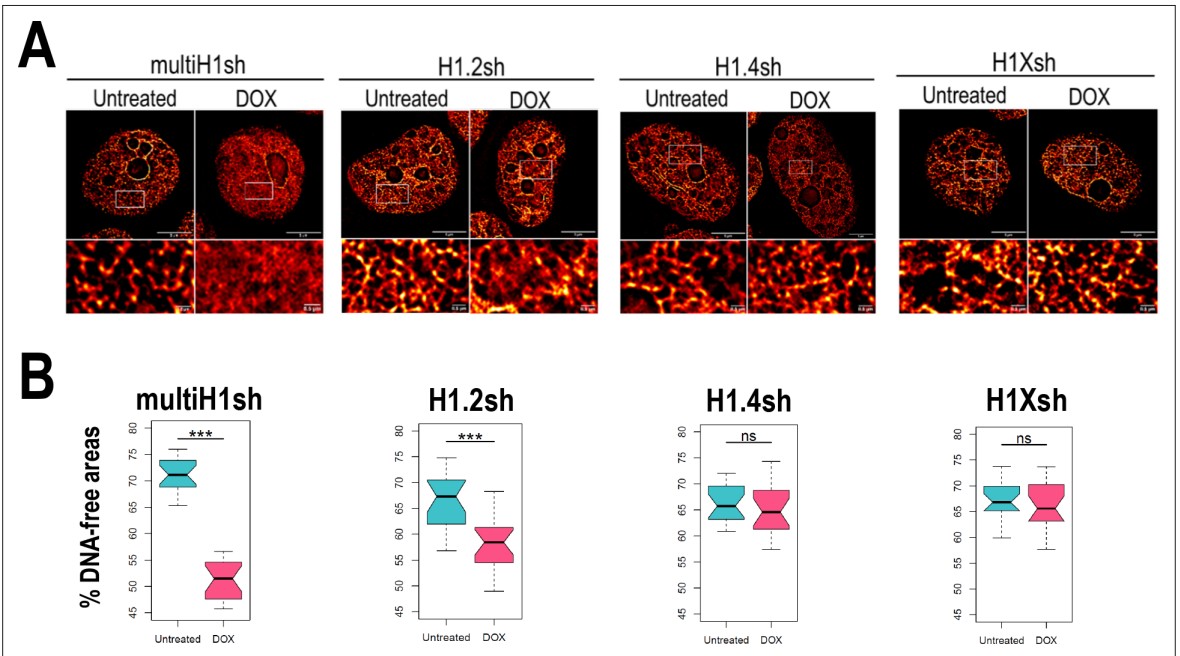

**Figure 4.** Chromatin structural changes upon H1 depletion. (**A**) Representative super-resolution radial fluctuation (SRRF) images of DNA staining in the different H1 knock-downs (KD) conditions indicated (multi-H1, H1.2, H1.4, and H1X Dox-inducible shRNAs). In the bottom panels, a zoom-in inset is shown to appreciate DNA pattern in both Untreated and Dox conditions. Scale bar: 5 μm (full nucleus) and 500 nm (zoom-in). (**B**) DNA-free areas percentage quantification in the different H1 KDs. n = 20 cells/condition were quantified and the boxplot were constructed with the 20 average values in each condition. Statistical differences between Untreated and Dox-treated conditions are supported by paired *t*-test. (***) p-value <0.001; (ns/non-significant) p-value >0.05. Additional representative images and full quantification are shown in *Figure 4—figure supplement 1*.

The online version of this article includes the following figure supplement(s) for figure 4:

**Figure supplement 1.** Super-resolution imaging of DNA upon different H1 knock-downs (KD) conditions.

abundant at regions that are only classified as NADs (NADs only) than NAD/LAD genomic regions, contrary to low-GC H1s.

## Single or combined H1.2 depletion leads to chromatin decompaction

Due to the intrinsic link of chromatin compartmentalization and genome structure and the differential H1 variants abundance within those compartments, we examined how H1 depletion affects chromatin organization. Using super-resolution imaging of DNA we were able to visualize the chromatin structure (*Figure 4—figure supplement 1A*). We performed super-resolution imaging of DNA under different H1 (*Figure 4A*, *Figure 4—figure supplement 1B*). Multi-H1 knock-downs (KD; i.e. simultaneous depletion of H1.2 and H1.4, see *Izquierdo-Bouldstridge et al., 2017*) led to a general disruption of chromatin organization compared to control conditions. This disruption was also evident, albeit to a lesser extent, upon single depletion of H1.2. In contrast, single depletion of H1.4 or H1X did not appear to produce changes in chromatin architecture at the level studied. To quantify chromatin structural changes upon H1 depletion, we used DNA-free areas analysis, as reported elsewhere (*Martin et al., 2021*; *Neguembor et al., 2021*). This method relies on the fact that DNA signal accumulates in densely packed areas. This leads to the appearance of areas with no DNA signal or low-density DNA signal (referred to as DNA-free areas). Considering this, upon DNA decompaction, a decrease in the percentage of DNA-free areas is expected. We quantified DNA-free areas under the different H1 KD conditions (*Figure 4B*, *Figure 4—figure supplement 1C*). Thus, the percentage of DNA-free areas was strongly reduced upon multi-H1 KD, supporting DNA decompaction. Similarly, H1.2 depletion also led to a decreased percentage of DNA-free areas, although the reduction was minor compared to multi-H1 KD. Depletion of H1.4 or H1X did not lead to significant changes in % DNA-free areas. In conclusion, combined depletion of H1.2 and H1.4 but also single depletion of H1.2 have an impact on chromatin structure, leading to a general chromatin decompaction not seen when depleting other H1 variants.

## The nuclear distribution of H1 variants is mainly conserved among different cell lines with a full H1 complement but altered in cells with silenced H1.3 and H1.5

Total H1 content and the contribution of H1 variants to total H1 are known to vary among cell types but little is known about the comparative nuclear or genomic distribution of the H1 variants among cell types. To address H1 variants heterogeneity in human cells, we firstly investigated the protein content of H1 variants in different cell lines, most of which had a tumoral origin (*Figure 5—figure supplement 1A–D*). As previously reported, H1.2 and H1.4 were present in all tested cell lines (*Lennox and Cohen, 1983*; *Meergans et al., 1997*; *Parseghian and Hamkalo, 2001*; *Piña and Suau, 1987*). Notably, H1X was also universally expressed. While H1.0 was only absent in HeLa cells, H1.3 and/or H1.5 proteins were not expressed in several cell lines, that is Hela, HepG2, HCT-116, HT-29, 293T, SK-MEL-173, IGR-39 (*Figure 5—figure supplement 1A–D*), and MDA-MB-231 (data not shown). Specifically, we consistently found a simultaneous lack of both H1.3 and H1.5 (i.e. HCT-116, HT-29, SK-MEL-173, and IGR-39). Interestingly, cell lines lacking H1.3 and H1.5 tend to have increased H1.0 levels compared to other cell lines tested. We also evaluated H1 variants mRNA expression levels by reverse-transcriptase-quantitative PCR (polymerase chain reaction) (RT-qPCR) (*Figure 5—figure supplement 1E*). H1.2 was the most expressed at mRNA level in all cell lines. Simultaneous absence of H1.3 and H1.5 seen at the protein level (*Figure 5—figure supplement 1A–D*) was also evident at the transcriptional level (*Figure 5—figure supplement 1E*). For this reason, we next explored whether the concomitant repression of H1.3 and H1.5 may be mediated by DNA methylation. Analysis of NCBI-60 cell lines panel showed that H1.0, H1.1, H1.5, and H1.3 expression levels exhibit a negative correlation with gene methylation status and they were not expressed in all cell lines (*Figure 5—figure supplement 2A*). This observation indicates that in some cell lines, expression of these variants could be repressed by DNA methylation. Moreover, gene methylation data from cancer patients (The Cancer Genome Atlas, TCGA) revealed that methylation of H1 variant genes varied between cancers originating from different tissues (*Figure 5—figure supplement 2B*). H1.2, H1.4, and H1X genes were unmethylated in the three datasets analyzed, supporting their universal expression in human cells. On the other hand, gene methylation of the other H1 variants was variable.

We next explored whether transcriptional repression of H1 variants was reversed by inhibition of DNA methylation. In those cell lines lacking H1.3+H1.5, a huge mRNA upregulation of these variants occurred upon 5-aza-2'-deoxycytidine (aza) treatment (*Figure 5—figure supplement 2C*). H1.0 expression was also upregulated in HeLa cells (which lack H1.0 protein) but to a lesser extent. Moreover, H1.1 expression was also upregulated upon aza treatment in all cell lines. Notably, H1.1 is not expressed basally in most cell lines (*Figure 5—figure supplement 1E*). In summary, our analyses support that H1.3 and H1.5 could be repressed by DNA methylation in a subset of cell lines.

To evaluate differential and common distribution patterns of H1 variants among different cell lines, we performed immunofluorescence of six endogenous H1 variants in some of the cell lines in which we characterized H1 complement (*Figure 5A, B*). Importantly, H1.2/H1.3/H1.5 were universally enriched at the nuclear periphery, as observed in T47D. H1.0 and H1.4 were distributed throughout the nucleus. Lastly, H1X was also distributed throughout the entire nucleus, but the intensity of its nucleolar enrichment was variable between cell lines. Importantly, similar profiles were observed in cell lines with a non-tumoral origin (*Figure 5—figure supplement 3*).

As both H1.3 and H1.5 seemed to be universally enriched at LADs or peripheral chromatin, we investigated whether in cell lines lacking these two variants, re-distribution of the remaining H1 proteins to the nuclear periphery occurs. To do so, we performed H1 variants immunofluorescence in cell lines lacking H1.3 and H1.5 (*Figure 5B*). In this subset of cell lines, H1.2 was also enriched at the nuclear periphery. Interestingly, H1.4 and H1.0 appeared to have a more peripheral distribution compared to cell lines expressing all H1 variants evaluated here. Indeed, we quantified H1.0 and H1.4 radial distribution (as exemplified in *Figure 1B*) and confirmed that those cell lines with a compromised H1 repertoire presented a more peripheral distribution of both H1.4 and H1.0, compared to those cell lines expressing all H1 variants (*Figure 5C*, *Figure 5—figure supplement 4A and B*). These results suggest that H1.0 and H1.4 balance H1 content at the nuclear periphery when H1.3 and H1.5 are absent, indicating that H1 levels are important to maintain peripheral chromatin and, presumably, LADs.

We next focused our research on histone H1X. H1X was distributed throughout the whole nucleus in a punctuated pattern, with a variable nucleolar enrichment between different cell lines (*Figure 5A, B*). Co-immunostaining of H1X with the nucleolar marker NPM1 confirmed H1X nucleolar enrichment in both tumoral and non-tumoral cell lines (*Figure 5D*). Among the cell lines tested, T47D, MCF-7, SK-MEL-147, HCT-116, 293T, and IMR-90 showed the most prominent H1X nucleolar enrichment. Notably, in MCF-7 breast cancer cells, H1X formed a layer at the nucleolar rim, adjacent to NPM1, similar to what was observed in T47D cells. Nevertheless, it is essential to note that H1X was still present at nucleoli in all cell lines tested, making it the H1 variant most associated with nucleoli, where other H1 variants are underrepresented.

Non-nucleolar H1X coincides with less-stained DNA regions, suggesting their enrichment at less compact chromatin. To further study H1X distribution, we performed H1X ChIP-Seq in several of the cell lines analyzed (i.e. SK-MEL-147, MCF-7, SK-N-SH, HeLa, and HCT-116 and previously reported T47D). We used G-bands segmentation to compare the H1X ChIP-Seq abundance in the mentioned cell lines. H1X showed a strong correlation with %GC content in all cell lines, being highly enriched at high-GC G-bands (*Figure 5E*). Moreover, analysis demonstrates G-bands utility as epigenetic units to directly compare H1 variants binding profiles (*Serna-Pujol et al., 2021*).

Altogether, analysis of H1 variants distribution in different cell lines showed some universal features for certain variants. Concretely, H1.2, H1.3, and H1.5 are enriched toward the nuclear periphery in all cell lines tested. Interestingly, in cell lines lacking H1.3 and H1.5, H1.4 and H1.0 adopt a more peripheral distribution, suggesting a compensatory behavior. For its part, H1X is universally enriched at high-GC regions and is the only H1 variant with evident presence within the nucleoli, although H1X nucleolar enrichment is variable among cell lines.

## Discussion

Tethering of chromatin to scaffold structures, such as the nuclear lamina or the nucleolus, regulates genome conformation and ultimately, its function. Despite being highly abundant proteins in the nucleus, distribution of histone H1 variants within nuclear domains has not been explored. We have recently reported that H1 variants exhibit differential genomic distributions in T47D breast cancer cells. H1.0, H1.2, H1.3, and H1.5 are enriched at low-GC regions and B compartment while H1.4

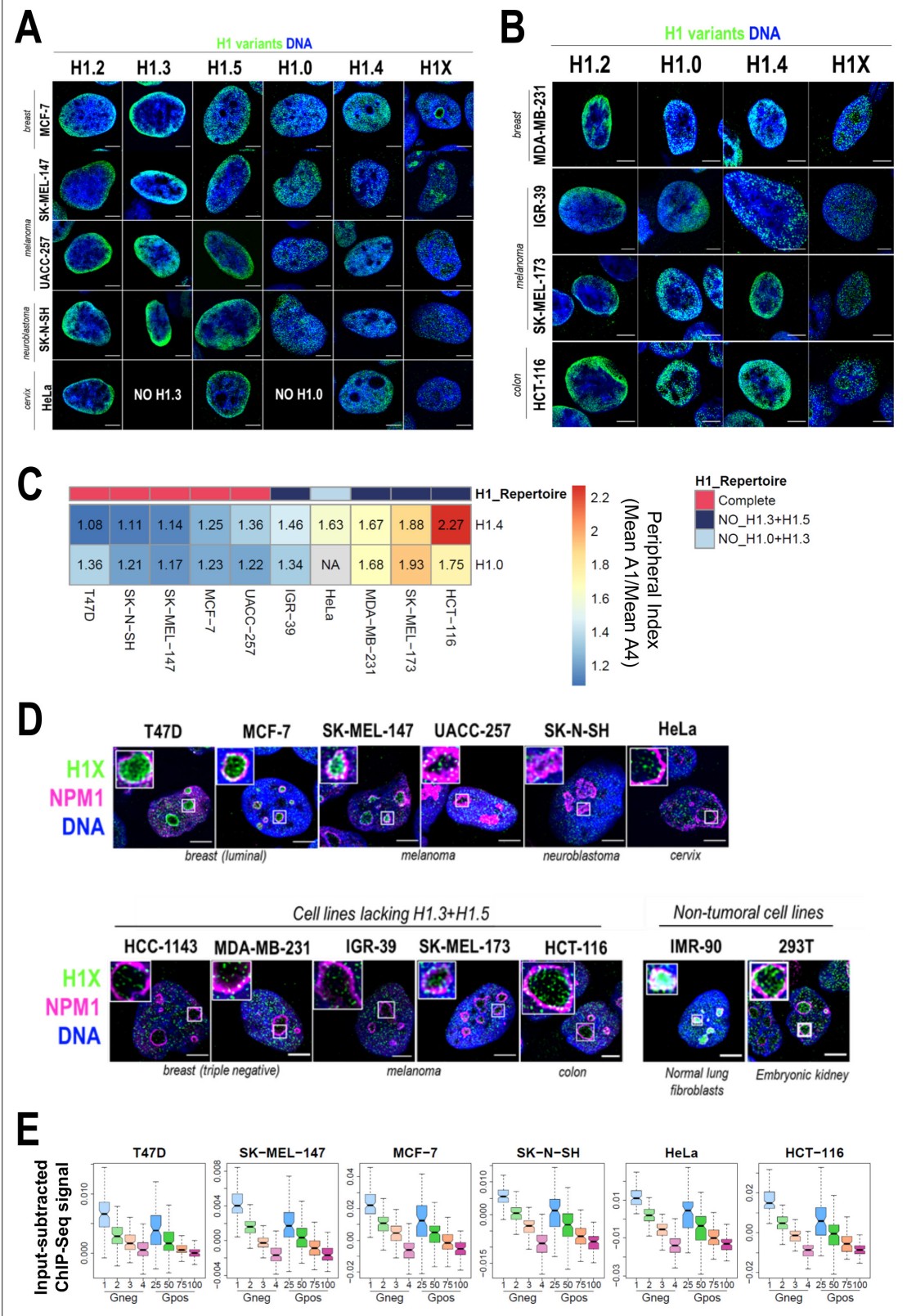

**Figure 5.** Nuclear distribution of H1 variants across multiple human cell lines. (**A**) Immunofluorescence analysis of H1 variants (green) with DNA staining (blue) in different cancer cell lines. Merged images are shown. H1.3 and H1.0 grids in HeLa cells are empty, as HeLa cells do not express these variants. Tumoral origin of the cell lines is indicated. Scale bar: 5 μm. (**B**) Immunofluorescence analysis of H1 variants (green) with DNA staining (blue) in cell lines lacking H1.3 and H1.5. Merged images are shown. Tumoral origin of the cell lines is indicated. Scale bar: 5 μm. (**C**) H1.4 and H1.0 show a more

*Figure 5 continued on next page*

*Figure 5 continued*

peripheral distribution in cell lines with a compromised H1 repertoire. Numbers correspond to peripheral index value in each cell line and color coded as indicated. Each nucleus was divided into four equivalent sections A1–A4 and immunofluorescence signal of H1.4 or H1.0 was quantified. Peripheral index was defined as the ratio between average value in A1 peripheral section and A4 central section. (**D**) Immunofluorescence of H1X (green), nucleolar marker Nucleophosmin (NPM1; magenta), and DNA staining (blue). Merged images are shown. Insets show a zoom-in of a single nucleolus. Bottom panel includes cell lines lacking H1.3 and H1.5. Cell line origin is indicated. Scale bar: 5 µm. (**E**) Boxplots show the H1X Input-subtracted ChIP-Seq signal at eight groups of Giemsa bands in six different cancer cell lines. G-bands groups were defined according to *Serna-Pujol et al., 2021* (see Materials and methods).

The online version of this article includes the following source data and figure supplement(s) for figure 5:

**Figure supplement 1.** H1 protein and mRNA complement across cell lines.

**Figure supplement 1—source data 1.** PDF containing original scans of the relevant Western blot analysis shown in *Figure 5—figure supplement 1A, C* with highlighted bands.

**Figure supplement 1—source data 2.** Original file for the Western blot analysis in *Figure 5—figure supplement 1A*, left panels (H1X).

**Figure supplement 1—source data 3.** Original file for the Western blot analysis in *Figure 5—figure supplement 1A*, left panels (H1.0).

**Figure supplement 1—source data 4.** Original file for the Western blot analysis in *Figure 5—figure supplement 1A*, left panels (H1.2).

**Figure supplement 1—source data 5.** Original file for the Western blot analysis in *Figure 5—figure supplement 1A*, left panels (H1.3).

**Figure supplement 1—source data 6.** Original file for the Western blot analysis in *Figure 5—figure supplement 1A*, left panels (H1.4).

**Figure supplement 1—source data 7.** Original file for the Western blot analysis in *Figure 5—figure supplement 1A*, left panels (H1.5).

**Figure supplement 1—source data 8.** Original file for the Western blot analysis in *Figure 5—figure supplement 1A*, left panels (H3).

**Figure supplement 1—source data 9.** Original file for the Western blot analysis in *Figure 5—figure supplement 1A*, right panels (H1X).

**Figure supplement 1—source data 10.** Original file for the Western blot analysis in *Figure 5—figure supplement 1A*, right panels (H1.0).

**Figure supplement 1—source data 11.** Original file for the Western blot analysis in *Figure 5—figure supplement 1A*, right panels (H1.2).

**Figure supplement 1—source data 12.** Original file for the Western blot analysis in *Figure 5—figure supplement 1A*, right panels (H1.3).

**Figure supplement 1—source data 13.** Original file for the Western blot analysis in *Figure 5—figure supplement 1A*, right panels (H1.4).

**Figure supplement 1—source data 14.** Original file for the Western blot analysis in *Figure 5—figure supplement 1A*, right panels (H1.5).

**Figure supplement 1—source data 15.** Original file for the Western blot analysis in *Figure 5—figure supplement 1A*, right panels (H3).

**Figure supplement 1—source data 16.** Original file for the Western blot analysis in *Figure 5—figure supplement 1C*, left panels (H1X).

**Figure supplement 1—source data 17.** Original file for the Western blot analysis in *Figure 5—figure supplement 1C*, left panels (H1.0).

**Figure supplement 1—source data 18.** Original file for the Western blot analysis in *Figure 5—figure supplement 1C*, left panels (H1.2).

**Figure supplement 1—source data 19.** Original file for the Western blot analysis in *Figure 5—figure supplement 1C*, left panels (H1.3).

**Figure supplement 1—source data 20.** Original file for the Western blot analysis in *Figure 5—figure supplement 1C*, left panels (H1.4).

**Figure supplement 1—source data 21.** Original file for the Western blot analysis in *Figure 5—figure supplement 1C*, left panels (H1.5).

**Figure supplement 1—source data 22.** Original file for the Western blot analysis in *Figure 5—figure supplement 1C*, left panels (H3).

**Figure supplement 1—source data 23.** Original file for the Western blot analysis in *Figure 5—figure supplement 1C*, left panels (H4).

**Figure supplement 1—source data 24.** Original file for the Western blot analysis in *Figure 5—figure supplement 1C*, right panels (H1X).

**Figure supplement 1—source data 25.** Original file for the Western blot analysis in *Figure 5—figure supplement 1C*, right panels (H1.0).

**Figure supplement 1—source data 26.** Original file for the Western blot analysis in *Figure 5—figure supplement 1C* right panels (H1.2).

**Figure supplement 1—source data 27.** Original file for the Western blot analysis in *Figure 5—figure supplement 1C*, right panels (H1.3).

**Figure supplement 1—source data 28.** Original file for the Western blot analysis in *Figure 5—figure supplement 1C*, right panels (H1.4).

**Figure supplement 1—source data 29.** Original file for the Western blot analysis in *Figure 5—figure supplement 1C*, right panels (H1.5).

**Figure supplement 1—source data 30.** Original file for the Western blot analysis in *Figure 5—figure supplement 1C*, right panels (H3).

**Figure supplement 1—source data 31.** Original file for the Western blot analysis in *Figure 5—figure supplement 1C*, right panels (H4).

**Figure supplement 2.** H1 variants expression regulation by DNA methylation.

**Figure supplement 3.** H1 variants nuclear distribution in non-tumoral cell lines.

**Figure supplement 4.** Nuclear radial distribution of H1.4 and H1.0 across cell lines.

**Figure supplement 5.** Cell lines lacking H1.3 and H1.5 show high basal expression of repetitive elements in comparison with cell lines with a complete H1 somatic repertoire.

and H1X are more abundant within high-GC and A compartment regions (*Serna-Pujol et al., 2022*). Moreover, a combined depletion of H1.2 and H1.4 leads to chromatin decompaction at the level of TADs (*Serna-Pujol et al., 2022*), demonstrating that H1 proteins are involved in maintaining genome structure. In this study, we profiled the differential nuclear distribution of six somatic H1 variants in T47D cells and other human cells lines, through imaging techniques, including super-resolution microscopy. We provide here the first systematic comparison of H1 variants distribution in multiple human cell lines.

In T47D cells, H1.2, H1.3, and H1.5 and to a lesser extent H1.0 are enriched toward nuclear periphery. On the other hand, H1X and H1.4 are distributed throughout the nucleus with H1X being highly enriched in nucleoli (*Figure 1A*). Super-resolution imaging of H1 variants reinforced these differential profiles and revealed that H1.2/H1.3/H1.5/H1.0 coincide more with DNA pattern compared to H1.4 and H1X (*Figure 2A*), confirming their segregation in two groups denoted by ChIP-Seq data. Both immunofluorescence and ChIP-Seq are performed on fixed cells and H1 are known to be highly mobile proteins. Thus, although our results demonstrate this variant-specific preferential distribution, it is unlikely that H1 variants are unable to dynamically bind other chromatin types.

Emerging evidence from super-resolution microscopy indicates that nucleosomes are grouped in heterogenous nanodomains termed 'clutches' (*Ricci et al., 2015*). Moreover, TADs represent structural chromatin folding units at the sub-megabase scale (*Dixon et al., 2012*; *Nora et al., 2012*; *Sexton et al., 2012*). We have shown that H1 variants form spatially separated nanodomains throughout the nucleus, visualized as a 'punctuate' signal by super-resolution imaging, but with the aforementioned differential variant-specific local enrichments. Similarly, super-resolution imaging of core histone H2B also present this clustered pattern in human fibroblasts (*Ricci et al., 2015*). It is important to mention that nucleosome clutches were originally defined using STORM technique (*Ricci et al., 2015*), whose resolution is higher than the one achieved by SRRF. For that reason, we favor the idea that nanodomains formed by H1 variants would be more equivalent to TADs or sub-TADs rather than to nucleosome clutches. In fact, shifts on the ChIP-Seq H1 variants distribution tend to coincide with TAD borders and H1 variants are more homogenous within the same TAD than between TADs (*Serna-Pujol et al., 2021*). This observation also highlights the relationship between H1 distribution and the structural properties of chromatin.

By conventional immunofluorescence, preferential localization of H1.0 with other 'low-GC' H1 variants over H1.4/H1X was observed (*Figure 2—figure supplement 1*). However, this preferential co-localization was not evidenced at the super-resolution level and all H1s co-localized similarly with H1.0 (*Figure 2B*). This observation may suggest that domains spatially arranged at the 3D level are homogeneously marked by a certain H1 variant and not by random H1 variants. Nevertheless, in a cell population those nanodomains could be marked by different H1 variants. This could explain why at single-cell level we lost preferential co-localization of 'low-GC' H1s with H1.0 (compared to H1.4/H1X) while by ChIP-Seq data, mega-base domains of 'low-GC' H1 variants coincide. If those nanodomains were homogeneously marked by the same H1 variant in the cell population, we would observe differential enrichments between H1 variants belonging to the same GC cluster, even at the mega-base level. This apparent intra-population ambiguity may be indicating a structural role of 'low-GC' H1 variants and emphasizes the existing partial redundancy among certain H1 variants.

H1.2, H1.3, and H1.5 are highly enriched at the nuclear periphery in T47D cells but also in all cell lines analyzed (*Figures 1 and 5A, B*). Super-resolution microscopy revealed that these H1 variants form an adjacent layer to lamina and highly co-localize with H3K9me2 (*Figure 3A–C*). H3K9me2 is not only a universal component of LADs, but also it is indispensable for peripheral heterochromatin anchoring to the nuclear lamina (*Poleshko et al., 2019*). Universal H1.2, H1.3, and H1.5 enrichment at nuclear periphery directly point to these H1 variants as conserved components of LADs, as has been described for H3K9me2. Furthermore, these H1 variants could be postulated as potential orchestrating factors for chromatin tethering to the lamina. Proper chromatin–lamina interactions are crucial to maintain chromatin dynamics (*Briand and Collas, 2020*; *Chandra et al., 2015*; *Chang et al., 2022*; *Zheng et al., 2018*). LADs detachment through Lamin B1 KO in human cells led to abnormal segregation of chromosome territories and A/B compartments, as well as global chromatin decompaction (*Chang et al., 2022*). Actually, we have shown that H1.2 depletion in T47D cells also led to a global chromatin decompaction (*Figure 4*).

Interactions between the nuclear lamina and LADs are disrupted at early stages of mitosis and re-established upon mitotic exit. In general, mitosis involves large structural reorganization of chromatin (*Imakaev et al., 2012*) that is accompanied by eviction of multiple chromatin factors from DNA (*Martínez-Balbás et al., 1995*). On the other hand, factors that persist attached to chromatin, including multiple histone variants and histone modifications (*Wang and Higgins, 2013*), are suggested to act as spatial 'bookmarks'. This is the case of H3K9me2, which is reported to safeguard positional information of LADs through mitosis, through a phospho-methyl switch (H3K9me2S10p) (*Poleshko et al., 2019*). Indeed, we have found that interphasic 'low-GC' H1 variants persist more attached to chromatin during mitosis, compared to 'high-GC' ones (*Figure 2E*, *Figure 2—figure supplements 2 and 3*). Moreover, H1.3 and H1.5, which are highly associated to LADs in interphase, persist in the peripheral layer of mitotic chromosomes, showing an analogous profile to the one reported for H3K9me2 (*Poleshko et al., 2019*). H1.2 attachment to mitotic chromatin is regulated by phosphorylation at early mitotic stages, but interestingly, H1.2 layer re-associates to the forming lamina upon mitotic exit. The strong similarities observed for H1.2, H1.3, and H1.5 with H3K9me2 in both interphase and mitosis suggest that these linker histones may also act as 3D positional 'bookmarks' of LADs.

We found that H1 variants may localize to nucleoli. Concretely, H1X and some phosphorylated H1s present different nucleolar distribution patterns (*Figure 3F*). H1X nucleolar enrichment has already been described in previous works (*Mayor et al., 2015*; *Stoldt et al., 2007*). In addition, other histone variants have been found at nucleoli, including testis-restricted H1T linker histone (*Tani et al., 2016*) or certain core histone variants (*Jiang et al., 2021*; *Long et al., 2019*; *Pang et al., 2020*). Importantly, nucleolar localization of H1X persists after inhibition of RNApol I by ActD (*Figure 3—figure supplement 1D*), which was also reported previously (*Stoldt et al., 2007*). These observations might suggest a more structural role of H1X in nucleoli rather than a more functional or regulatory one. On the contrary, H1.2-pT165 and H1.4-pT146 seem to execute a functional role, as the nucleolar distribution of these post-translationally modified H1 variants depends on functional nucleoli, with H1.4-pT146 being presumably associated to RNA pol I active transcription. The nucleolus is a membraneless organelle formed through liquid–liquid phase separation driven by multivalent interactions of its components (*Lafontaine et al., 2021*). Several molecular features are known drivers for phase separation, including highly intrinsically disordered regions (*Shin and Brangwynne, 2017*). Indeed, nucleolar proteome and specially proteins localized to the nucleolar rim are extremely disordered (*Stenström et al., 2020*). We found that nucleolar H1X is enriched, although not limited, at the nucleolar rim, adjacent to the inner side of NPM1 layer (*Figure 3E*). Histone H1 proteins have a well-known highly disordered structure and have been shown to phase separate in vitro (*Gibson et al., 2019*; *Shakya et al., 2020*; *Turner et al., 2018*). However, the functional relevance of H1 variants as promotors of phase separation in living cells has not been explored.

Proteomic studies in four human cell lines demonstrated that almost 1/3 of the candidate H1.0-binding proteins localized to nucleolus and were related to nucleolar functionality (*Kalashnikova et al., 2013*). Remarkably, the experiments were performed by pull-down of exogenous, chimeric HaloTag-H1.0 protein. Importantly, direct H1.0 nucleolar localization or rRNA metabolism alterations upon H1.0 depletion were not reported. In contrast, our results show that H1.0 is depleted from nucleoli in all cell lines analyzed (*Figure 5A, B*). We also checked that H1.0 does not redistribute to nucleoli upon H1X depletion in T47D cells (data not shown). Nevertheless, we cannot discard that H1.0 interacts with nucleolus-related proteins, as it can be enriched at perinucleolar heterochromatin or NADs, as observed in T47D cells.

We have explored how H1 variants depletion affect chromatin structure through super-resolution imaging of DNA (*Figure 4*). In T47D cells, combined depletion of H1.2 and H1.4 (i.e. multi-H1 KD) caused a global chromatin decompaction. This is in agreement with previously generated ATAC-Seq experiments in these cells, which pointed to a genome-wide gain of chromatin accessibility. Accordingly, Hi-C data analysis in multi-H1 KD cells also showed more de-compacted TAD structures (*Serna-Pujol et al., 2022*). In mice, multiple H1 variants deficiency has been also associated to chromatin decompaction (*Willcockson et al., 2021*; *Yusufova et al., 2021*). However, the differential contribution of individual H1 variants to chromatin structure has not been explored before. Interestingly, analysis of T47D single KDs revealed that single H1.2 depletion also led to chromatin decompaction, but not as pronounced as multi-H1 KD. On the contrary, single depletion of H1.4 or H1X did not cause a significant alteration of chromatin structure. These observations suggest that the structural defects

cannot be explained just for the total H1 reduction. Indeed, both H1.2 and H1.4 proteins contribution to total H1 content is the same in T47D, estimated to be 23–24% in each case (*Sancho et al., 2008*). Thus, H1 variant-specific functionality, related to their differential genomic distribution seems to play a role. These results could support the putative structural function of H1.2 (and maybe also for the rest of 'low-GC' variants). In multi-H1 KD cells, total H1 content is reduced ≈30% and chromatin decompaction is more drastic compared to single H1.2 depletion. Due to the fact that H1.2 and H1.4 occupy different genomic regions, the more drastic effects on decompaction in multi-H1 KD cells seem to be due, at least in part, to the additive depletion of two H1 variants with non-redundant functions. In the whole, super-resolution microscopy of DNA enables us to decipher global compaction changes upon several H1 KDs conditions and revealed that H1 variants have specific roles in shaping genome architecture. Moreover, both the total H1 reduction but also the H1 variant repertoire have an impact on the global chromatin compaction homeostasis.

H1 complement is known to be heterogeneous among cell types. We have observed that H1X is expressed in all cell lines tested, as it was previously reported with H1.2 and H1.4 (*Millán-Ariño et al., 2016*). Interestingly, the simultaneous lack of H1.3 and H1.5 is found recurrently and seems to be mediated by DNA methylation (*Figure 5—figure supplements 1 and 2*). On the contrary, whether H1 variants are universally distributed or display cell-type-specific patterns is not well understood. H1 variants are thought to be specifically distributed among different cell lines, but this presumption comes from combining various pieces of evidence from different publications (*Cao et al., 2013*; *Izzo et al., 2013*; *Li et al., 2012*; *Millán-Ariño et al., 2014*; *Torres et al., 2016*). They mostly address the analysis of a single H1 variant in a particular model or cell line. Moreover, a comparative study of the distribution of a single H1 variant in different cell models has not been performed so far. In the whole, the direct comparison of these studies is biased by the different origin of the data and the varied methodologies used, which in many cases involve the over-expression of H1 variants.

We performed the first systematic analysis of six endogenous variants in different cancer cell lines. Our data unveil, for the first time, universal nuclear patterns exhibited by specific H1 variants. Immunofluorescence experiments revealed that H1.2, H1.3, and H1.5 are universally enriched toward nuclear periphery. H1.0 and H1.4 are distributed throughout the whole nucleus but they show a more peripheral distribution in a subset of cell lines lacking H1.3 and H1.5 (*Figure 5*). H1X is also distributed throughout the whole nucleus in all cell lines tested with a variable relative nucleolar enrichment. H1.0 and H1.4 re-distribution when H1.3 and H1.5 proteins are absent contrast to the behavior observed in T47D multi-H1 KD. Upon multi-H1 KD, H1 variants distribution is overall robust, with no significant

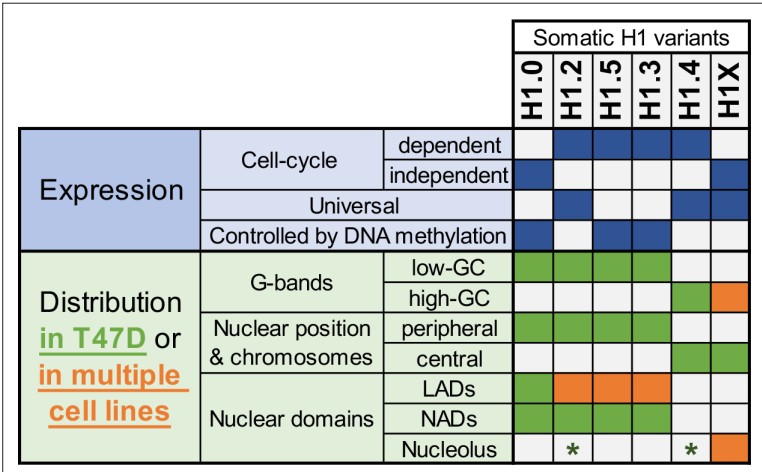

| | | | Somatic H1 variants | | | | | |
| | | | H1.0 | H1.2 | H1.5 | H1.3 | H1.4 | H1X |
|---|---|---|---|---|---|---|---|---|
| **Expression** | Cell-cycle | dependent | | ■ | ■ | ■ | ■ | |
| | | independent | ■ | | | | | ■ |
| | Universal | | | ■ | | | ■ | |
| | Controlled by DNA methylation | | | | ■ | ■ | | |
| **Distribution in T47D or in multiple cell lines** | G-bands | low-GC | ● | ● | ● | ● | | |
| | | high-GC | | | | | ● | ● |
| | Nuclear position & chromosomes | peripheral | | ● | ● | ● | | |
| | | central | ● | | | | ● | |
| | Nuclear domains | LADs | ● | ● | ● | ● | | |
| | | NADs | ● | ● | ● | ● | | |
| | | Nucleolus | | * | | | * | ● |

**Figure 6.** Summary of H1 variants expression and distribution specificities. The table summarizes the main aspects of H1 variants expression (blue; top part) and distribution (green/orange; bottom part). Regarding distribution characteristics (from microscopy and ChIP-Seq experiments), green boxes indicate features observed in T47D breast cancer cells. If the association has been universally found in multiple human cell lines, it is highlighted in orange. Green asterisks indicate that, in T47D, phosphorylated H1.2 and H1.4 have been found to be enriched at nucleoli. Additional notes: Concomitant absence of H1.3 + H1.5 has been found in several cell lines, which also exhibit a redistribution of H1.4 and H1.0 to the nuclear periphery.

changes at the nuclear level (*Serna-Pujol et al., 2022*). These represent two different H1-compromised scenarios. In the first one, H1.3 and H1.5 silencing is intrinsically linked to the cell line identity. On the contrary, multi-H1 KD cells represent an inducible H1.2 and H1.4 depletion, abnormal in T47D cells. Our results suggest that compensatory mechanisms between different H1 variants, in terms of distribution, may be limited when perturbing H1 levels but achieved when H1 repertoire is 'naturally' compromised. Regarding the functional outcomes of H1-restriction, multi-H1 KD cells are characterized by a robust interferon response, triggered by the expression of repetitive elements (*Izquierdo-Bouldstridge et al., 2017*). Notably, RT-qPCR experiments indicated that cells lacking H1.3 and H1.5 exhibit a high basal expression of the interferon signature and some repeats (*Figure 5—figure supplement 5*). Whether the concurrent loss of H1.3 and H1.5 represents an acquired adaptive mechanism in certain cancer cells, as well as the implications of the immune response triggered by H1 loss are interesting subjects for study.

In summary, our findings highlight the differential distribution of H1 variants within nuclear domains and their variant-specific role on chromatin (*Figure 6*). Moreover, we showed that H1 variants present a potentially more uniform distribution among cell lines than previously anticipated, particularly for certain variants.

# Materials and methods
## Cell lines, culturing conditions, and H1 variants KD
Breast cancer T47D-MTVL derivative cell lines, which carry one stably integrated copy of luciferase reporter gene driven by the MMTV promoter, were grown in RPMI 1640 medium, supplemented with 10% fetal bovine serum (FBS), 2 mM L-glutamine, 100 U/ml penicillin, and 100 µg/ml streptomycin, as described previously. SK-MEL-147, SK-MEL-173, UACC-257, SK-N-SH, HeLa, HCT-116, HT-29, CaCo-2, HepG2, 293T, NT2-D1, and IMR-90 cell lines were grown in DMEM GlutaMax medium, supplemented with 10% FBS and 1% penicillin/streptomycin. IGR-39, SK-MEL-28, and WM266-4 cell lines were grown in DMEM GlutaMax medium, supplemented with 10% FBS, 1% penicillin/streptomycin, and 1% HEPES (4-(2-hydroxyethyl)-1-piperazineethanesulfonic acid). MCF-7 cell line was grown in MEM medium containing 10% FBS, 1% penicillin/streptomycin, 1% non-essential aminoacids, 1% sodium pyruvate, and 1% L-glutamine. MDA-MB-231 cell line was grown in Dulbecco's modified Eagle medium (DMEM)/F-12 medium containing 10% FBS, 1% penicillin/streptomycin, and 1% L-glutamine. All cell lines were grown at 37°C with 5% $CO_2$. Cell lines were tested for absence of mycoplasma contamination.

Doxycycline (Dox)-inducible shRNA H1 KD were described in previous works (*Izquierdo-Bouldstridge et al., 2017*; *Mayor et al., 2015*; *Sancho et al., 2008*). Concretely, T47D H1.4sh (*Izquierdo-Bouldstridge et al., 2017*), T47D H1.2sh (*Sancho et al., 2008*), and T47D H1Xsh (*Mayor et al., 2015*) cell lines were used to analyze single H1 depletion. The T47D-MTVL multi-H1 shRNA cell line (*Izquierdo-Bouldstridge et al., 2017*) was used as a model for H1 depletion. In multi-H1 KD, combined depletion of H1.2 and H1.4 proteins occurs, although it reduces the expression of several H1 transcripts. A derivative cell line containing a Randomsh RNA was used as a control (*Sancho et al., 2008*). For details on tagged-HA H1.0 stable expression in T47D cell line, see *Millán-Ariño et al., 2014*.

## Drug treatments
shRNA expression was induced with 6 days treatment of Dox, in which cells were passaged on day 3. Dox (Sigma) was added at 2.5 µg/ml. 5-aza-2'-deoxycytidine (aza) was added at 5 µM for 3 days, in which medium was replaced at day 2 by fresh aza-containing medium. ActD was added at 50 ng/ml for 24 hr. To study the distribution of H1 variants in mitosis by immunofluorescence in T47D, we performed Thymidine-Nocodazole synchronization, in order to increase the percentage of mitotic cells in the sample.

## Immunofluorescence
Cells were directly grown on glass coverslips (0.17 mm thickness, 1.5 H high performance; Marienfeld Superior) placed in 24-well plates. Cells were fixed with 4% paraformaldehyde (20 min; RT), permeabilized with methanol (10 min RT) and blocked with 5% bovine serum albumin (5% BSA diluted in

phosphate-buffered saline [PBS]-Triton 0.1%). Primary antibodies of interest were incubated overnight at 4°C. Secondary antibodies conjugated to Alexa fluorophores were incubated 1 hr RT in the dark. The following conjugated secondary antibodies (Invitrogen) were used: goat anti-rabbit IgG H+L (Alexa-488 or -647); donkey anti-mouse IgG H+L (Alexa-555, -561, -633, or -647). After incubation, samples were washed with PBS-T (x3) and nuclei were stained with Hoechst (25 µg/ml diluted in 5% BSA-PBS-T; 1 hr RT in the dark). Five PBS-T washes and a final MiliQ water were performed. Coverslips were mounted using Pro-long glass (Invitrogen). Preparations were maintained 24–48 hr in the dark at RT and then stored at 4°C up to image acquisition.

## Image acquisition

All images were acquired in a *Dragonfly 505* multimodal spinning-disk confocal microscope (Andor Technologies, Inc), using a ×100/NA-1.49 Apochromat oil immersion objective, a sCMOS Andor Sona 4.2B-11 camera and Fusion acquisition software. Pinhole diameter 40 µm was used for confocal imaging. Laser excitation was done sequentially. Alexa-633 and -647 were excited by 647 nm laser, Alexa-555 and -561 were excited by 561 nm laser, Alexa-488 was excited by 488 nm laser, and Hoechst was excited by 405 nm laser. Exposure time and laser intensity were adapted in each case, ensuring the absence of saturating pixels. 16-bit images were acquired. Confocal 3D images were taken as Z-stacks with 0.11 µm intervals, with a voxel size of 51 × 51 × 110 nm. GPU-assisted deconvolution (Regularized Richardson-Lucy, 16 iterations) was applied after acquisition using the Fusion software. Deconvolved images are shown and representative confocal images show a single focal plane, unless indicated in the figure legend.

SRRF algorithm (*Culley et al., 2018*; *Gustafsson et al., 2016*) was applied using the SRRF-Stream+ module (Andor) operated from the Fusion software. SRRF was performed for (1) co-localization experiments and (2) chromatin structure evaluation at extended resolution. For SRRF co-localization experiments a single Z confocal plane 1024 × 1024 was imaged. When evaluating H1 variants co-localization with DNA, H1 variants were labeled with Alexa-488 (except for H1.0, which was labeled with Alexa-561) and Hoechst was used for DNA staining. For H1 variants co-localization with H1.0, H1 variants were labeled with Alexa-488 and H1.0 with Alexa-561. Images frames were also acquired sequentially for each channel. The following parameters were used for all conditions: 1× ring radius, 6× radiality magnification (i.e. each pixel is magnified in an array of 6 × 6 sub-pixels), 500 frames. Exposure time 250 ms and 18% laser intensity were used for 405 nm channel (Hoechst imaging). Exposure time 180–200 ms and 8–12% laser intensity were used for 488 or 561 nm channels. Under these conditions, pixel size corresponds to 8.5 nm (in *x,y*).

For evaluating DNA structure upon H1 KDs, SRRF imaging of DNA (Hoechst) was used. SRRF was performed on 1024 × 1024 wide-field images with the following parameters: exposure time 150 ms, 10% 405 nm laser intensity, 1× ring radius, 6× radiality magnification, 1000 frames. Under these conditions, pixel size corresponds to 8.5 nm (in *x,y*).

## Image analysis

Image analysis was performed in ImageJ software. Data were post-processed and plotted in R or Excel. Fluorescence intensity quantification was done by generating masks for each nucleus using Hoechst signal as reference and computing the mean intensity of the proteins of interest. Alternatively, Corrected Total Cell Fluorescence formula was used and calculated as: Integrated density – area of selected nucleus × mean fluorescence of background readings. Line signal intensity profile plots were created using Plot Profile tool. Analysis of ring intensity distribution was done with a macro available at https://github.com/MolecularImagingPlatformIBMB/ringIntensityDistribution ( *Rebollo, 2019* ) with minor modifications. Every nucleus is partitioned into four concentric rings with equal areas that converge toward the center of the nucleus. Subsequently, the signal intensity density of the specific interest is assessed for each ring and adjusted to the total intensity density of the nucleus. This approach enables the comparison of the intensity distribution of a target protein among nuclei varying in shape and size. Pearson's correlation coefficient (*r*) was calculated using JaCoP Plugin. Calculation was done from a unique Z central plane in single nucleolus and after channel thresholding. Co-localization in SRRF images was calculated with an in-house macro. Briefly, after pre-processing steps, it creates a mask of each channel and calculates the intersection between both masks of interest (in % of intersected pixels). To compare chromatin structure through DNA super-resolution in different

H1 KD conditions, we used DNA-free areas method, which has been used elsewhere to assess DNA compaction (*Martin et al., 2021*; *Neguembor et al., 2021*). We constructed an interactive macro that allowed us to sample each nucleus using user-defined regions of interest (ROI) and then automatically calculates the free-DNA areas per ROI. For each nucleus, a number of ROIs (200 × 200 px) were drawn as to cover all the nuclear area, excluding nucleolus. Per each ROI, auto-local thresholding using the Phansalkar filtering algorithm was applied to the Hoechst channel. Based on this filtering, percentage of DNA-free areas was calculated. Biological replicates of imaging experiments were performed and figures show representative cells. Number of cells used for quantification are indicated in the corresponding figure legends.

## Histones extraction

For isolation of total histones, cell pellets were resuspended in 1 ml of hypotonic solution [10 mM Tris–HCl (pH 8.0), 1 mM KCl, 1.5 mM MgCl$_2$, 1 mM PMSF (phenylmethylsulfonyl fluoride), 1 mM DTT (dithiothreitol)] and incubated on ice for 30 min. The nuclei were pelleted at 10,000 × *g* for 10 min at 4°C. Sulfuric acid (0.2 M) was added to the pellet to extract the histones on ice for 30 min. The solution was centrifuged at 16,000 × *g* for 10 min at 4°C. TCA (trichloroacetic acid) was added to the supernatant in order to precipitate histones. After >1 hr ice-incubation precipitate was centrifuged (16,000 × *g* 10 min at 4°C). Precipitate was washed with acetone and finally resuspended in water. Protein concentration was determined by Micro BCA protein assay (Thermo Scientific) and immunoblot was performed.

## Immunoblot

Histone samples were exposed to sodium dodecyl sulfate–polyacrylamide gel electrophoresis (14%), transferred to a PVDF membrane, blocked with Odyssey blocking buffer (LI-COR Biosciences) or 5% non-fat milk for 1 hr, and incubated with primary antibodies overnight at 4°C as well as with secondary antibodies conjugated to fluorescence (IRDye 680 goat anti-rabbit IgG or IRDye 800 goat anti-mouse IgG, Li-Cor) for 1 hr at room temperature. Bands were visualized in an Odyssey Infrared Imaging System (Li-Cor). Coomassie staining or histone H3/histone H4 immunoblotting were used as loading controls. H1 protein content was quantified from Coomassie staining of histone extracts using ImageJ software. H1 variants can be visualized in three consecutive bands (35–32 kDa, corresponding to H1.3 + H1.4 + H1.5, H1.2, and H1.0, respectively), as indicated in *Figure 5—figure supplement 1A, C*. H1X cannot be quantified from Coomassie staining. The relative intensity of each H1 band was corrected by H4 band (loading control) and expressed as a percentage of total H1 content.

## Antibodies

Specific antibodies recognizing human H1 variants used for immunofluorescence, immunoblot, and ChIP-Seq were: anti-H1.0/H5 clone 3H9 (Millipore, 05-629-I), anti-H1.2 (abcam, ab4086), anti-H1.3 (abcam, ab203948), anti-H1.4 (Invitrogen, 702876), anti-H1.5 (Invitrogen, 711912), and anti-H1X (abcam, ab31972). Other antibodies used in immunofluorescence and/or immunoblot were: H1.2-pT165 (H1.2 phosphorylated in Thr-165; Millipore 06-1370), H1.4-pT146 (H1.4 phosphorylated in Thr-146; ab3596), Lamin A (ab8980), H3K9me2 (ab1220), NPM1 (ab10530), H3 (ab1791), H4 (ab10158), and HA (ab9110). For immunoblots in *Figure 5—figure supplement 1*, H1.3 (ab24174) antibody was used. In *Figure 5—figure supplement 1A*, H1.0 immunoblot was performed with H1.0 (ab11079) antibody. Of note, anti-H1.4pT146 immunogen was a synthetic peptide derived from within residues 100–200 of human H1.4, phosphorylated at T146. However, this antibody could also recognize phosphor-T146 in H1.2, H1.3 (both 88% sequence identity with immunogen). Importantly, source of H1 variants antibodies limit co-immunostaining studies. All H1 variants antibodies are raised in rabbit, except for H1.0 antibody, which is raised in mouse. Notably, performance and specificity of H1 variant antibodies have been extensively validated in our previous publication (*Serna-Pujol et al., 2022*). However, some additional immunofluorescence validations are reported in *Figure 1—figure supplement 1*.

## RNA extraction and RT-qPCR

Total RNA was extracted using the High Pure RNA Isolation Kit (Roche). Then, cDNA was generated from 100 ng of RNA using the Superscript First Strand Synthesis System (Invitrogen). Gene products

were analyzed by qPCR, using SYBR Green Master Mix (Invitrogen) and specific oligonucleotides in a QuantStudio 5 machine (Applied Biosystems). To determine the H1 variants expression contribution to total mRNA H1, each value was corrected by GAPDH and by the corresponding cell line genomic DNA amplification of each primer pair. Then, the relative expression of each H1 variant was expressed as a percentage of the total somatic H1 mRNA expression. Specific qPCR oligonucleotide sequences are listed in *Supplementary file 1*.

## ChIP and ChIP-Seq

All H1 ChIP-Seq experiments were performed and analyzed as previously detailed (*Serna-Pujol et al., 2022*). ChIP-Seq replicates of H1.0, H1.2, H1.4, H1.5, and H1X in T47D cells from our previous publication (*Serna-Pujol et al., 2022*) are accessible through GEO Series accession numbers GSE156036 and GSE166645. Input-subtracted ChIP-Seq signal was evaluated in genome segments of interest using BEDTools (*Quinlan and Hall, 2010*). Genome-wide GC content, Giemsa bands (G-bands) coordinates at 850 bands per haploid sequence (bphs) resolution and chromosomes coordinates were obtained from the UCSC human genome database (*Karolchik, 2004*; *Navarro Gonzalez et al., 2021*). G-bands were classified in eight groups as detailed in *Serna-Pujol et al., 2021*: G-positive (Gpos25 to Gpos100, according to its intensity in Giemsa staining), and G-negative (unstained), which were further divided into four groups according to their GC content (Gneg1 to Gneg4, from high to low-GC content). For Figures construction, a single ChIP-Seq replicate was used in each case, although analogous results were obtained with biological replicates included in the listed GEO accession numbers.

## Public data on H1 variants expression and H1 gene methylation

H1 variants expression data and gene methylation in the NCI-60 cell lines panel were available at CellMiner (https://discover.nci.nih.gov/cellminer/; *Reinhold et al., 2019*). Gene methylation data from Illumina 450K methylation BeadChip were expressed as $\beta$-values normalized to a value between 0 (unmethylated) and 1 (methylated). $\beta$-Values from all identifiers (i.e. different probes) corresponding to the same H1 gene were considered and an average $\beta$-value was calculated per each H1 variant. Gene methylation in cancer patients from TCGA datasets was available at https://www.cbioportal.org/.

## Acknowledgements

We acknowledge Núria Serna for bioinformatics support on genomics data analysis, Pau Homs for assistance on some immunofluorescence quantifications, Stefany Montúfar and Núria Pell for assistance on immunoblot analysis of cell lines, and Carles Bonet for help with cell line conservation and laboratory managing. We acknowledge Generalitat de Catalunya and the European Social Fund for an AGAUR-FI predoctoral fellowship [to MS-P].

## Additional information

### Funding

| Funder | Grant reference number | Author |
|---|---|---|
| Spanish Ministry of Science and Innovation | PID2020-112783GB-C21/AEI/10.13039/501100011033 | Albert Jordan |

The funders had no role in study design, data collection and interpretation, or the decision to submit the work for publication.

### Author contributions

Monica Salinas-Pena, Conceptualization, Data curation, Formal analysis, Validation, Visualization; Elena Rebollo, Data curation, Software, Methodology; Albert Jordan, Conceptualization, Supervision, Funding acquisition, Writing - original draft, Writing - review and editing

### Author ORCIDs

Albert Jordan ⓘ http://orcid.org/0000-0002-3970-8693

Reviewer #1 (Public Review): https://doi.org/10.7554/eLife.91306.3.sa1
Reviewer #2 (Public Review): https://doi.org/10.7554/eLife.91306.3.sa2
Reviewer #3 (Public Review): https://doi.org/10.7554/eLife.91306.3.sa3
Author Response https://doi.org/10.7554/eLife.91306.3.sa4

## Additional files

### Supplementary files
• Supplementary file 1. Oligonucleotides for semiquantitative PCR. Forward (F) and reverse (R) oligonucleotides for the indicated genes are shown.
• MDAR checklist

### Data availability
ChIP-Seq data of H1X in five different cancer cell lines have been deposited in NCBI's Gene Expression Omnibus (GEO) and is accessible through GEO Series accession number GSE236678. ChIP-Seq of H1.3 in T47D cells is available through the accession number GSE236878.

The following previously published datasets were used:

| Author(s) | Year | Dataset title | Dataset URL | Database and Identifier |
|---|---|---|---|---|
| Jordan A, Salinas M, Serna N | 2023 | Genome-wide profiling of linker histone variant H1X in five different cancer cell lines | https://www.ncbi.nlm.nih.gov/geo/query/acc.cgi?acc=GSE236678 | NCBI Gene Expression Omnibus, GSE236678 |
| Jordan A, Salinas M, Serna N | 2023 | Genome-wide profiling of linker histone variant H1.3 in T47D-MTVL cancer cell line | https://www.ncbi.nlm.nih.gov/geo/query/acc.cgi?acc=GSE236878 | NCBI Gene Expression Omnibus, GSE236878 |

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
