## [Editor Report · eLife assessment]

This manuscript is an **important** advance in the study of Histone H1s, finding distinct distributions of various H1 variants in the genome. The controls presented by the authors provide **convincing** evidence to demonstrate a heterogenous distribution of H1 which might reflect functional regulation of chromatin accessibility by linker histones. This work will be of interest to the genome organization field, and could additionally provide a framework for understanding H1 mis-regulation observed in cancer cells.

---

## [Referee Report · Reviewer #1 (Public Review)]

In this manuscript, authors have performed extensive imaging analysis of six human histone H1 variants, their enrichment and localization, their differential dynamics during interphase and mitosis, and their association with lamina-associated domains (LADs) or nucleolus-associated domains. The manuscript is well-written with high-quality confocal and super-resolution images. Various interesting observations are made on distribution patterns of H1 variants. H1.2, H1.3, and H1.5 are shown to be universally enriched at the nuclear periphery whereas H1.4 and H1X are found to be distributed throughout the nucleus. Interestingly, H1X was the only H1 variant found to be abundant in nucleoli. Depletion of H1 variants has been shown to affect chromatin structure in a variant-specific manner, with H1.2 knock-down resulting in global chromatin decompaction. Overall, the study presents several interesting insights on H1 variants conducted in a large number of cell lines.

---

## [Referee Report · Reviewer #2 (Public Review)]

Summary:

The manuscript by Salinas-Pena et. al examines the distribution of a subgroup of histone H1 variants primarily with the use of high-resolution microscopy. The authors find that while some H1s have a universal distribution pattern, some display a preference for discrete regions within the nuclear landscape namely, the periphery, the center, or the nucleolus. They also show using that the various H1s within a cell did not colocalize significantly with each other, rather, they occupy discrete 'nanodomains' throughout the nucleus which is visualized as a punctate signal.

The authors present evidence towards a long-standing question in the field regarding the spatial distribution of the different H1 variants. Since reliable, specific antibodies toward the variants were unavailable, this question was unable to elicit a definitive answer. This study uses more recently available antibodies against endogenous H1s to put together a systematic and comprehensive view of a group of H1 variant distribution inside a nucleus and ties it with previously generated genome wide data to demonstrate localization and some functional heterogeneity.

Strengths of the study.

(1) First systematic, high-resolution view of H1 variants providing a significant advance towards the long hypothesized functional differences between H1 variants.

(2) The use of endogenous antibodies allows the authors to bypass the need to use tagged proteins or overexpression strategies to study H1 distribution.

(3) The availability of genome wide H1 distribution data for the variants using the endogenous H1 antibodies to strengthen the presented visual data.

Weakness of the study.

One of the major reasons for slow progress in deciphering variant specific function has been the dearth of quality, specific, antibodies. This study is heavily dependent on the antibody function and its ability to accurately report on the distribution. The authors have cited previous validations of the antibodies used using H1 knockdown, immunoblotting and ChIP-seq. For the scope of this study, the controls are adequate.

Impact:

This study sets the stage for an exciting avenue of H1 study where variant-specific cellular functions can be explored which has otherwise been severely understudied.

---

## [Referee Report · Reviewer #3 (Public Review)]

Summary:

This paper uses indirect immunofluorescence, superresolution fluorescence microscopy, and X-ChIP to demonstrate radial distribution profiles of all histone H1 somatic variants with the exception of histone H1.1. The results support earlier work from chromatin immunoprecipitation experiments that revealed biases for active versus repressed states of chromatin. The previous studies provided some support for the subtle sequence variation found primarily within the C-terminus of histone H1 variants conferred preferences in the type of DNA (e.g. methylated DNA) or chromatin bound. The current study significantly strengthens that argument. Importantly, this was shown across multiple cell lines and reveals conserved properties of localization of histone H1 variants.

Strengths:

The strength of the manuscript is the combined use of quantitative analysis of indirect immunofluorescence and X-ChIP. The results generally support the polar organization of the genome and a corresponding distribution of histone H1 variants that reflect this polar organization. AT-rich chromatin is positioned near the lamina and is found to be enriched in H1.2, H1.3, and H1.5. H1.4 and H1.X were more biased towards the GC-rich intranuclear chromatin.

There is emerging functional evidence for variant-specific properties to histone H1 subtypes. This work provides an important building block in understanding how different histone H1 variants may have specific functional consequences. The histone H1 variant that is most abundant in most cell types, H1.2, was found to decrease the area of the immunofluorescent slice that was chromatin-free when depleted, suggesting a more important role in global chromatin organization.

Weaknesses:

While histone H1 variants may show biases in their distributions, it is unlikely that these are more than biases. That is, it is unlikely that specific H1 variants are unable to bind to nucleosomes in regions where they are depleted. Fluorescence recovery after photobleaching experiments have demonstrated differences in binding affinity but the capacity to bind a range of chromatin structures, including highly acetylated chromatin, for histone H1 variants. Thus, it is critical in assessing this data to have accurate quantitative information on the relative abundance of the different histone variants amongst the cell lines tested here. The paper relies upon quantification by immunoblotting.

Another uncertainty in both the ChIP and immunofluorescence datasets is the accessibility of the epitope. This weakness is highlighted by the apparent loss of H1.2 and H1.4 in mitotic chromosomes that is revealed to be false by the detection of the phosphorylated species. The distributions relative to the surface of chromosomes in mitosis and the depletion of H1.2, H1.3, and H1.5 from the central regions of interphase nuclei reveals an unusual dissipation of the staining that is suggestive of antibody accessibility problems. The overall image quality of the immunofluorescence images is poor, further complicating analysis.

---

## [Author Response]

The following is the authors’ response to the original reviews.

We appreciate the insightful feedback provided by the editors and reviewers who have recognized the novelty of our study. We have mapped the spatial distribution of six endogenous somatic histone H1 variants within the nuclei of several human cell lines using specific antibodies, which strongly suggest functional differences between variants. We are submitting a revised version of the manuscript to accommodate the reviewers comments and recommendations.

**Reviewer #1 (Recommendations For The Authors):**
Minor Comments:(1) In Figure 1C, since H1.4 is uniformly distributed among the four sections (A1-A4), its levels are not expected to be significant among the four sections as depicted. Even the violin plots shown do not seem to be significantly different from each other. This requires an explanation.

We agree with this reviewer that significant differences of H1.4 abundance within areas A1 to A4 seem to not exist, either looking at the images or the data violin plots, as discussed in the manuscript. Nonetheless, statistical testing gave this as significant, due to small differences and the elevated sample N of the analysis. It is clear that H1.4 does not show a relevant peripheral enrichment as shown for the other variants.

(2) At the end, it would be better to include a figure panel depicting chart/table/pictorial representation, depicting the summary of the work done with respect to all the histone variants, as there are several histone H1 variants studied under different conditions and contexts.

A table summarizing the location and characteristics of the different H1 variants has been included in the manuscript (Figure 6).

**Reviewer #2 (Recommendations For The Authors):**
(1) The authors may consider adding controls for the specificity of the antibodies used for the studies. While the antibodies used here are commercial, it does not guarantee the quality for immunofluorescence, especially considering their unreliability in the past. The authors may consider including peptide/ recombinant protein-based adsorption controls in addition to knockdown or knockout controls. Having these data will strengthen the exciting observations presented in this MS and significantly increase the impact of the presented findings.

We totally agree with the reviewers that the use of commercially available antibodies does not guarantee their quality and specificity. As this issue was crucial for our studies, we extensively assayed performance and specificity of the antibodies, using different approaches. The validations were shown in our previous publications where these antibodies where successfully used for ChIP-seq (Serna-Pujol et al. 2022 NAR 50:3892; Salinas-Pena et al. 2024 NAR doi 10.1093/nar/gkae014). In summary, performance of H1.0 (05-629l, Millipore), H1.2 (ab4086, abcam), H1.4 (702876; Invitrogen), H1.5 (711912, Invitrogen) and H1X (ab31972; abcam) antibodies was tested by Western-Blot, ChIP and proteomic analyses (all the results are included in Supplem. Figure 1 in Serna-Pujol et al. 2022 NAR 50:3892). Concretely, we tested specificity using inducible KDs for the depletion of each of the somatic H1 variants in T47D. We also checked that the antibodies did not recognize additional H1 variants using recombinant proteins or cell lines naturally lacking some of the variants. All the experiments confirmed that antibodies were variant-specific. In addition, when the corresponding epitope was absent, the antibodies did not gain new cross-reactivity with other variants. More recently, validation of the specificicity of the H1.3 antibody (ab203948) was performed following the same experimental approaches described for the rest of antibodies (Supplem. Figure 1 in Salinas-Pena et al. 2024 NAR doi 10.1093/nar/gkae014).

(2) Histone H1 is overexpressed in several cancers. While the authors do not use an overexpression strategy, the cells used in this study are all cancer cell lines. The study would benefit greatly if some of the findings- primarily regarding the spatial distribution of the H1 were to reproduce in non-tumorigenic, diploid cells.

We have also studied and discussed the spatial distribution of H1 variants in nontumorogenic cell lines 293T and IMR-90, and we have added this in the revised manuscript (Figure 5D and Figure 5-figure supplement 3). The nuclear radiality of H1.4 in 293T cells is also shown (Figure 5-figure supplement 4A).

**Reviewer #3 (Recommendations For The Authors):**
This is an interesting paper that provides convincing evidence of distinct distributions to individual histone H1 variants. There are several aspects of the study that leave me unconvinced that the study accurately captures histone H1 variant distributions.(1) Antibody accessibility: (see PMID: 32505195). One means to address this is to express a fluorescent protein-tagged version of histone H1 and demonstrate that the antibody can detect that tagged version of histone H1 independent of its location in the nucleus. In general, these FP-tagged H1s show a much more even distribution than what is observed here. Of course, that could reflect artifacts related to the fusion or the expression of the exogenous construct. However, even if all of the above are true, this will test the ability of the antibodies to recognize their epitopes in different chromatin environments. The fluorescent protein tag enables unambiguous knowledge of the presence or absence of the H1 histone.

We have used cells expressing HA-tagged H1.0 variant and performed immunofluorescence with HA and H1.0 antibody to investigate co-localization, to test whether an H1 antibodiy recognize all the tagged protein in different chromatin environments or irrespective of its location in the nucleus. A very high correlation between the two antibodies has been found (Figure 1-figure supplement 1B).

(2) At high concentrations, the fluorescence signal intensity can be quenched. For example, this is common with high-affinity histone H3 serine 10 phosphorylation antibodies in late interphase/prophase nuclei. The artifact can be minimized by serial dilution of the antibody and identifying the minimum usable concentration for immunofluorescence. While I am not certain that this is taking place here, the rate and manner that the intensity drops off from the periphery in the peripheral H1 variant distribution are very similar in appearance. There are biological explanations related to constraints on diffusion that one could imagine also explaining the data so I'm not stating that this must be an artefact. However, I am concerned that it might be. An improved staining may reveal the same result but more convincingly.

We have performed immunofluorescence with serial dilutions of the H1.3 antibody to show that peripheral distribution was not due to fluorescence signal intensity quenching (Figure 1figure supplement 1A).

(3) Histone H1 is highly mobile and there is some concern that they could reorganize during the relatively long period of time that it takes to fully fix the cells for both ChIP and immunofluorescence. This should be acknowledged in the manuscript.

We have added this reviewer’ concern in the Discussion section.

(4) The paper would benefit from a more rigorous quantification of histone H1 subtypes. Mass spectrometry would be ideal but more classical techniques such as 2D AU-SDS PAGE, HPLC, etc...would be an improvement over immunoblotting. The authors did not explain the quantification of the immunoblots and the assignment of relative contributions of H1 subtypes to the individual coommassie bands in the Image J section of methods, which is referred to as the method of quantification in the immunoblotting methods.

We have further explained how the relative quantification of H1 variants in different cell lines was performed (Methods section). We agree that more sophisticated mass spectrometrybased quantification is desirable and we are collaborating to do this using internal H1 peptide controls (Parallel Reaction Monitoring), but this is out of the scope of this manuscript as the observed patterns of distribution of H1 variants do not depend on mild differences in variants abundance. Only the absence of H1.3 and H1.5 in some cell lines alters the distribution of other variants.

Additional author responses to the Public Review comments made by some Reviewer:

(1) Respect to the functional significance of the results presented here, we want to stress that as a consequence of the differential distribution and abundance of H1 variants among cell types, depletion of different variants has different consequences. For example, H1.2 depletion but not others has a great impact on chromatin compaction. Besides, cell lines lacking H1.3/H1.5 expression present a basal up-regulation of some Interferon stimulated genes (ISGs) and particular repetive elements, as it was previously described upon induced depletion of H1.2/H1.4 in a breast cancer cell line or in pancreatic adenocarcinomas with lower levels of replication-dependent H1 variants (Izquierdo et al. 2017 NAR 45:11622). So, our results reinforce the existing link between H1 content and immune signature. We have added this data in the revised manuscript (Figure 5-figure supplement 5).

Moreover, we also analyzed the chromatin structural changes upon combined depletion of H1.2 and H1.4. Combined H1.2/H1.4 depletion triggers a global chromatin decompaction, which supports previous observations from ATAC-Seq and Hi-C experiments in these cells (Izquierdo et al. 2017 NAR 45:11622; Serna-Pujol et al. 2022 NAR 50:3892). Although H1 content is more compromised in these cells (30% total H1 reduction) compared to single H1 KDs, the phenotype observed could not be recapitulated when other H1 KD combinations, in which total H1 content was reduced similarly, were investigated (Izquierdo et al. 2017 NAR 45:11622), supporting that the deleterious defects were due to the non-redundant role of H1.2 and H1.4 proteins. Indeed, this manuscript supports this notion, as H1.2 and H1.4 show a different genomewide and nuclear distribution.

(2) Our immunofluorescence data, together with ChIP-seq data, do not discard binding of H1 variants to a great variety of chromatin, but show enrichment or preferential binding to certain regions or chromatin types. Our data on the interphase nuclei does not suggest at all any type of quenching or saturation. Obviously, detection with antibodies depends on epitope accessibility, just like all immunofluorescence data ever published, and we have acknowledged that post-translational modifications of H1 may occlude antibody accessibility as some phospho-H1 antibodies give distribution patterns different than total/unmodified H1 antibodies. Thus, we cannot exclude that specific modified-H1s exhibit particular distribution patterns that are not being recapitulated in our data. This represents another layer of complexity in H1 diversity and we agree that exploration of the repertoire of H1 PTMs and their functional roles are an interesting matter of study that needs to be addressed. Still, our data is highly relevant as it demonstrates for the first time the unique distribution patterns of H1 variants among multiple cell lines and it does not use overexpression of tagged H1 variants that in our experience may produce mislocalization of H1s.

(3) We do have investigated co-localization of H1 variants with HP1alpha protein and we have added this data in the revised version of this manuscript (Figure 1-figure supplement 1C-D).